# An allosteric site in the T-cell receptor Cβ domain plays a critical signalling role

Kannan Natarajan[1], Andrew C. McShan[2], Jiansheng Jiang[1], Vlad K. Kumirov[2], Rui Wang[1], Huaying Zhao[3], Peter Schuck[3], Mulualem E. Tilahun[1], Lisa F. Boyd[1], Jinfa Ying[4], Ad Bax[4], David H. Margulies[1],* & Nikolaos G. Sgourakis[2],*

The molecular mechanism through which the interaction of a clonotypic αβ T-cell receptor (TCR) with a peptide-loaded major histocompatibility complex (p/MHC) leads to T-cell activation is not yet fully understood. Here we exploit a high-affinity TCR (B4.2.3) to examine the structural changes that accompany binding to its p/MHC ligand (P18-I10/H2-D$^d$). In addition to conformational changes in complementarity-determining regions (CDRs) of the TCR seen in comparison of unliganded and bound X-ray structures, NMR characterization of the TCR β-chain dynamics reveals significant chemical shift effects in sites removed from the MHC-binding site. Remodelling of electrostatic interactions near the Cβ H3 helix at the membrane-proximal face of the TCR, a region implicated in interactions with the CD3 co-receptor, suggests a possible role for an allosteric mechanism in TCR signalling. The contribution of these TCR residues to signal transduction is supported by mutagenesis and T-cell functional assays.

[1] Molecular Biology Section, Laboratory of Immunology, National Institute of Allergy and Infectious Diseases, National Institutes of Health, Bethesda, Maryland 20892, USA. [2] Department of Chemistry and Biochemistry, University of California Santa Cruz, Santa Cruz, California 95064, USA. [3] Laboratory of Cellular Imaging and Macromolecular Biophysics, National Institute of Biomedical Imaging and Bioengineering, National Institutes of Health, Bethesda, Maryland 20892, USA. [4] Laboratory of Chemical Physics, National Institute of Diabetes and Digestive and Kidney Diseases, National Institutes of Health, Bethesda, Maryland 20892, USA. * These authors contributed equally to this work. Correspondence and requests for materials should be addressed to D.H.M. (email: dmargulies@niaid.nih.gov) or to N.G.S. (email: nsgourak@ucsc.edu).

A key step in T-cell-mediated adaptive immunity is the triggering of cell-surface αβ T-cell receptors (TCR) by peptide-loaded major histocompatibility complex (p/MHC) proteins on target antigen presenting cells[1,2]. TCR-α and -β polypeptide chains are encoded by genes assembled by recombinatorial assortment of V-J and V-D-J gene segments, respectively, and non-templated nucleotides added at junctions of rearrangement during T-cell ontogeny in the thymus. Encounter of particular clonally expressed TCR with cognate p/MHC ligand triggers a signalling cascade leading to a variety of cellular programmes including thymic selection, proliferation, cytokine production and differentiation into effector and memory T cells[3].

Whereas antigen specificity is dictated by the amino-terminal variable (V) domains of the αβ-receptor, signalling function is mediated by the non-covalently associated co-receptor CD3γε, δε and ζζ dimers, which bear cytoplasmic immunoreceptor tyrosine-based activation motifs (ITAMs)[4,5]. Ligand binding to the TCR/CD3 complex extracellularly initiates intracellular signalling through Src kinase-mediated phosphorylation of these ITAMs[6]. In addition to their signalling function, CD3 subunits are also required for stable cell-surface expression of the TCR/CD3 complex[7,8]. Mechanistic details concerning the transmission of signals from the extracellular domains of the TCR to the intracellular ITAMs are incomplete, and are the subject of considerable interest, the importance of which is highlighted by diseases associated with dysfunction of this cellular process[9], the immunosuppressant role of therapeutic antibodies targeting the TCR/CD3 complex[10] and the potential of synthetic TCRs towards immunotherapeutic applications[11,12]. Efforts to understand the molecular basis of TCR-mediated signalling have relied largely on biophysical, structural and functional approaches[13]. Binding of p/MHC to the TCR induces structural changes at the cytoplasmic face of the TCR/CD3 complex, as evidenced by the accessibility of a polyproline sequence in the CD3ε cytoplasmic tail[14], and the repositioning of Tyr residues within the CD3 cytoplasmic ITAMs from a relatively inaccessible membrane-associated form to a cytoplasmically oriented, kinase-accessible conformation[15]. However, the molecular mechanism by which p/MHC binding to the TCR is communicated to the associated CD3 subunits for signalling remains unknown.

To gain further insight into the dynamics of TCR/MHC interactions, we employ complementary biophysical methods to examine the high-affinity B4.2.3 TCR in both the liganded and unliganded states. X-ray structures indicate a large rearrangement of the complementarity-determining region 3 (CDR3) loops upon binding. In addition, chemical shift mapping utilizing complementary backbone amide and side-chain methyl NMR probes reveal several residues in the Cβ domain of the TCR, distant from the ligand-binding interface and close to a putative CD3-binding site, that show significant perturbations upon ligand binding. Finally, mutational and functional analyses suggest a critical role of these allosteric sites in signal transduction. These results indicate a dynamic activation mechanism, where p/MHC recognition by the CDRs triggers conformational remodelling of interactions near the Cβ H3 helix at the membrane-proximal face of the TCR.

## Results

**TCR binds to its pMHC ligand with high affinity.** The B4.2.3 T-cell hybridoma, derived from a BALB/c mouse immunized with P18-I10 (RGPGRAFVTI), is sensitive to picomolar concentrations of peptide presented by the MHC-I molecule, H2-D[d] (refs 16,17). To probe the affinity and kinetics of the interaction between the TCR and p/MHC, we first employed surface plasmon resonance (SPR) where immobilized P18-I10/H2-D[d] was offered graded concentrations of the B4.2.3 TCR. The measured affinity ($K_D$) is

$\sim 0.54\,\mu M$ (Fig. 1a), and no binding of the TCR was detected to H2-D[d] displaying the motif peptide (MTF) AGPARAAAL, a negative control (Fig. 1b). As TCR affinities for p/MHC ligands span a wide range of $K_D$ from 0.5 to $>100\,\mu M$ (ref. 18), this affinity is among the highest reported for a naturally occurring TCR.

To explore the TCR/MHC interaction in a complementary assay, we employed sedimentation velocity analytical ultracentrifugation, which permits assessment of the stoichiometry as well as the affinity of the interaction. The B4.2.3 TCR interacts with P18-I10/H2-D[d] strongly over a broad range of concentrations (0.1–20 μM of each component) displaying a characteristic concentration-dependent sedimentation coefficient distribution, indicative of a greater time average of molecules in complex as a function of concentration (Fig. 1c). The limiting sedimentation coefficient of $\sim 5S$ is consistent with a 1:1 stoichiometry, considering the size and shape of the individual components. The strong interaction of the TCR with the cognate P18-I10/H2-D[d] ligand contrasts sharply with mixtures of B4.2.3 TCR and MTF/H2-D[d] (Fig. 1d), both of which sediment at a concentration-independent velocity of 3.6S. To determine the binding affinity, we analysed the isotherm of signal weighted-average sedimentation coefficient ($s_w$) using the $s_w$ values determined from the $c(s)$ distributions. A simple 1:1 association model with affinity ($K_D$) of 0.23 μM fits the data well (95% confidence interval: 0.14–0.40 μM). Thus, the B4.2.3 recombinant TCR interacts with P18-I10/H2-D[d] with high affinity, in two distinct biophysical assays, consistent with the high peptide sensitivity of the T-cell hybridoma in functional assays[16,17].

**TCR uses plasticity in the CDR3 loops to recognize pMHC.** To elucidate the details of the ligand/receptor interaction, we determined the crystal structures of both free and p/MHC-bound states of the B4.2.3 TCR. The unliganded TCR crystallized in the P3$_1$ space group, with three heterodimers in the asymmetric unit (data collection and refinement statistics are provided in Table 1). The P18-I10/H2-D[d]/B4.2.3 complex formed crystals in the C2 space group and diffracted to 2.1 Å, revealing a structure with an overall orientation of the TCR on P18-I10/H2-D[d] that conforms to previously elucidated general principles[19–22] (Fig. 2a). The germline-encoded CDR1 and CDR2 loops of the TCR α-chain are positioned towards the C-terminal half of the H2-D[d] α2 helix, resulting in a diagonal orientation mode (Fig. 2b). The resulting crossing angle calculated as described[22] is 30°, within the range observed for the majority of stimulatory TCR/MHC complexes (22°–87°; ref. 22). The shape complementarity index[23] of the interface is 0.71 (1.0 is a perfect match), and is among the highest observed for any TCR/MHC complex[22]. A summary of the interactions between selected residues of the CDRs with the MHC α1 and α2 helices and peptide is shown in Fig. 2c. While α- and β-subunits contribute almost equally to the 1,771 Å[2] area of interaction between B4.2.3 and H2-D[d] (calculated without peptide), nine of ten hydrogen bonds at the interface are provided by the β-chain CDRs. Thus, this pMHC/TCR interface is dominated by Vβ interactions within the broad range observed for many different pMHC/TCR complexes[22].

Superposition of the α- and β-chains of the liganded B4.2.3 TCR with their unliganded counterparts reveals marked changes in the disposition of the CDR3α and CDR3β loops, with little alteration discernible in CDR1 and CDR2 or in the C domains (Supplementary Fig. 1a–d) as observed in other TCRs[22,24]. In the structures examined here, CDR3α, spanning Ala95 to Lys102, undergoes a large conformational change upon ligand binding, in particular a 9.1 Å displacement of the C$_\alpha$ atom of Asp99 of the first molecule in the asymmetric unit of the unliganded TCR compared to the same atom in the liganded TCR (Supplementary

Fig. 1b). The movement projects the CDR3α loop into the peptide-binding groove, and allows interactions of the Phe97 side chain with the peptide backbone of Gly2 and Gly4. This is illustrated by a 9.2 Å displacement of the $C_\zeta$ atom of the Phe97 side chain from the free to bound form. CDR3β, from Ser92 to Val99, is displaced inwards on ligand binding by 3.1 Å as measured at the $C_\alpha$ of His96. Thus, the X-ray structures indicate large movements of the CDR3α and -β loops upon p/MHC engagement that are critical for achieving a highly complementary interface. The extensive interface and hydrogen-bonding network between the α- and β-CDR domains with the P18-I10 peptide suggests an enthalpic compensation for the apparent entropic loss due to loop rearrangement and provide a structural basis for the measured high affinity of the interaction, as demonstrated both functionally and biochemically.

**Increased dynamics in regions of the TCR β-chain in solution.** To gain insight into the dynamics of the B4.2.3 TCR in solution in both its free and P18-I10/H2-D$^d$-bound states, we prepared TCR samples for NMR labelled at the β-chain alone by in vitro assembly of unlabelled α-chain with triple-labelled ($^2$H,$^{13}$C,$^{15}$N) or AILV side-chain methyl-labelled β-chain. The resulting proteins showed well-dispersed spectra, indicating stable, properly

conformed monomeric TCR, free of aggregation or degradation (Fig. 3a and Supplementary Fig. 2).

Previous studies have established strategies for obtaining NMR assignments of TCR backbone atoms using TROSY-based methods under extensive perdeuteration of the α and β-chains[25,26]. Here we employed a multipronged approach for assignment and cross-validation of both backbone amide and side-chain methyl resonances of isotopically labelled β-chain TCR samples (outlined in Supplementary Fig. 2). First, backbone amide chemical shifts in the 2D $^1$H-$^{15}$N TROSY-HSQC were assigned sequentially utilizing a combination of three-dimensional (3D) HNCA, 3D HN(CA)CB and 3D HNCO experiments recorded on a triple-labelled sample. Second, Ile $^{13}$Cδ1, Leu $^{13}$Cδ1/$^{13}$Cδ2, Val $^{13}$Cγ1/$^{13}$Cγ2 chemical shift assignments were obtained from 3D HMCM[CG]CBCA methyl out-and-back experiments[27] recorded on a selectively methyl-labelled sample[28] (Supplementary Fig. 3). Backbone amide assignments obtained using the J-correlated experiments were validated by acquiring amide-to-amide NOEs in TROSY-based 3D $H_N$-$NH_N$ nuclear Overhauser enhancement spectroscopy (NOESY) experiments, while side-chain methyl assignments were validated with NOE connectivities obtained from 3D $H_M$-$C_M H_M$ and 3D $C_M$-$C_M H_M$ SOFAST NOESY experiments[29]. Close comparison of the NOE crosspeak intensities to their corresponding distances in the X-ray structure further permitted stereospecific disambiguation of the geminal Leu

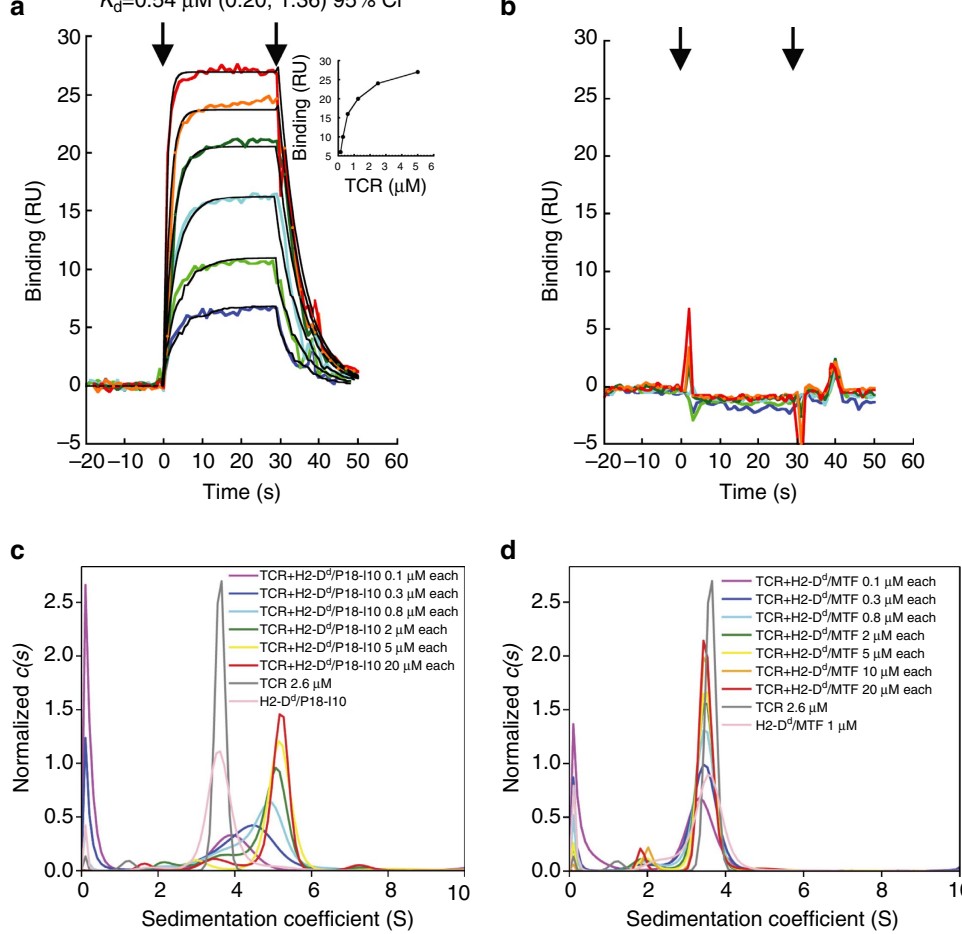

**Figure 1 | TCR binds pMHC with high affinity and peptide specificity.** (a,b) SPR analyses of B4.2.3 binding to P18-I10/H2-D$^d$ (a) or to MTF/H2-D$^d$ (b). B4.2.3 was offered to immobilized P18-I10/H2-D$^d$ complexes at concentrations of 0.16, 0.32, 0.62, 1.25, 2.5 and 5.0 μM and data were analysed as detailed in the Methods. Inset to a shows binding isotherm with indicated fitted parameters in the top of the plot. (c,d) Sedimentation velocity analytical ultracentrifugation (SV-AUC) analyses of interaction of B4.2.3 TCR with P18-I10/H2-D$^d$ (c) or with MTF/H2-D$^d$ (d). Centrifugation and analysis are described in the Methods.

**Table 1 | Data collection and refinement statistics.**

| | P18-I10/H2-D<sup>d</sup>/B4.2.3 | B4.2.3 TCR free |
|---|---|---|
| *Data collection* | | |
| Space group | C2 | P3₁ |
| Cell dimensions | | |
| *a, b, c* (Å) | 210.11, 51.32, 93.74 | 96.11, 96.11, 167.58 |
| *α, β, γ* (°) | 90.00, 97.14, 90.00 | 90.00, 90.00, 120.00 |
| Resolution (Å) | 29.8-2.1 (2.2-2.1)* | 48.1-3.00 (3.1-3.0)* |
| $R_{merge}$(%)<sup>†</sup> | 8.2 (33.2)* | 13.1 (48.2)* |
| $I/\sigma(I)$ | 16.3 (2.2)* | 12.6 (1.8) |
| Completeness (%) | 97.8 (80.1)* | 94.4 (65.3)* |
| Redundancy | 6.2 (2.6)* | 5.4 (3.8)* |
| Estimated twin fraction/law<sup>‡</sup> | 0.0/(none) | 0.458/(h,-h-k,-l) |
| | | |
| *Refinement* | | |
| Resolution (Å) | 29.8-2.1 (2.2-2.1)* | 48.0-3.0 (3.1-3.0)* |
| No. of reflections (unique) | 57,154 | 32,680 |
| $R_{work}$<sup>§</sup>/$R_{free}$<sup>‖</sup> (%) | 18.0 (23.2)*/22.0 (30.0)* | 24.0 (39.3)*/26.8 (44.1)* |
| No. of atoms | 7,027 | 10,062 |
| Wilson B-factor | 36.3 | 70.3 |
| Average B-factor | 52.0 | 91.3 |
| R.m.s deviations | | |
| Bond lengths (Å) | 0.002 | 0.002 |
| Bond angles (°) | 0.66 | 0.80 |
| Ramachandran favoured/outliers (%) | 96.0/0.8 | 96.0/0.2 |
| PDB code | 5IVX | 5IW1 |

*Asterisked numbers correspond to the last resolution shell.
†$R_{merge} = \Sigma_h \Sigma_i |I_i(h) - <I(h)>|/\Sigma_h \Sigma_i I_i(h)$, where $I_i(h)$ and $<I(h)>$ are the ith and mean measurement of the intensity of reflection *h*.
‡A pseudo-merohedral twin fraction was estimated by Xtriage in PHENIX.
§$R_{work} = \Sigma_h ||F_{obs}(h)| - |F_{calc}(h)||/\Sigma_h |F_{obs}(h)|$, where $F_{obs}(h)$ and $F_{calc}(h)$ are the observed and calculated structure factors, respectively. No $I/\sigma(I)$ cutoff was applied.
‖$R_{free}$ is the R value obtained for a test set of reflections consisting of a randomly selected 5% subset of the data set excluded from refinement.

$^{13}C\delta1/^{13}C\delta2$ and Val $^{13}C\gamma1/^{13}C\gamma2$ resonances. Finally, the combined backbone amide and side-chain methyl assignments were cross-validated using NOEs obtained from 3D $H_N$-$C_M H_M$ and 3D $C_M$-$NH_N$ SOFAST NOESY experiments[29]. This approach allowed us to achieve backbone amide assignments for 90% of non-Pro residues and side-chain methyl assignment of 100% Ileδ1, Leuδ1/δ2, Valγ1/γ2 and Alaβ methyls for β-chain labelled TCR. Examples of our sequential assignment and NOESY-based cross-validation strategy for two selected regions of the β-chain are presented in Supplementary Figs 4 and 5. Thus, with the addition of complete, stereospecific assignments of methyl groups to our backbone assignments we obtained a complementary network of probes towards mapping dynamics of the TCR β-chain in solution.

The backbone assignments and NOE connectivity patterns further confirm the structural features of the TCR β-chain, including the identification of all conserved Ig domain disulfide bonds in an oxidized form as evidenced by the $^{13}C_\alpha$ and $^{13}C_\beta$ chemical shifts of the four Cys residues (22, 90 in Vβ and 141, 202 in Cβ), as well as the observation of a single peak for all residues in the vicinity of the disulfides. Resonances from several residues within the β-chain CDR loops, CC′ loop and FG loop (Supplementary Fig. 6) were absent in the TROSY spectra, likely due to conformational exchange line-broadening, suggesting the sampling of alternative environments on a μs–ms timescale. This is consistent with the multiple crystallographic conformations observed for the same residues among the three TCR molecules in the asymmetric unit (Supplementary Fig. 1d).

The NMR backbone chemical shifts are highly sensitive probes of the local environment and secondary structure of the molecule. On the basis of the assignments of backbone $^{1}H$, $^{15}N^H$, $^{13}C_\alpha$, $^{13}C_\beta$ and $^{13}CO$ atoms, we used TALOS-N[30] to calculate the secondary structure index and found these predictions to be in excellent agreement with the DSSP[31] annotation of our X-ray structure of the free TCR (Fig. 3c,d). When provided with

near-complete assignments, the detail of structural information contained in the chemical shifts analysis is highlighted by the robust prediction of the boundaries for the H3 and H4 α-helices in the Cβ domain, as well as the $3_{10}$ helical segment (residues 113–115) located in the linker between Vβ and Cβ (shown as red bars in Fig. 3c, and coloured red in the diagram of Fig. 3d). The chemical shift-derived order parameter (RCI-S²), expressed in the range 0–1 with 0 indicating a random coil and 1 a fully ordered backbone structure, further reveals regions of increased disorder. As opposed to the more detailed model-free order parameters[32] derived from fitting $^{15}N$ relaxation rates and $^{15}N$-{$^{1}H$} NOE ratios that probe ps–ns timescale motions directly, this analysis is based on a statistical comparison of backbone chemical shifts with database values for random coils using an empirical formula parameterized by comparison to molecular dynamics simulations[33]. Among regions of the TCR β-chain (lower plot in Fig. 3c), the β₂–β₃ loop (residues 13–17) and the β₇–β₈ loop (residues 68–72), not directly involved in p/MHC binding, as well as the CDR1 and CDR3 loops in the Vβ domain, all show decreased order parameter values with correspondingly increased crystallographic B-factors. The most unstructured region of the β-chain of the TCR in solution is the FG loop located in Cβ (residues 206–225), with RCI-S² values systematically below 0.75 for most residues and as low as 0.5 for Gly218. In contrast, the CC′ loop (residues 157–166) shows only a minor reduction in backbone rigidity, with values in the range 0.8–0.9. However, the CC′ loop may still undergo dynamic motions at a longer time scale window (μs–ms), as suggested by the missing resonances for three amides in that region due to conformational exchange line-broadening or increased solvent exchange rates.

**β-chain residues involved in the α/β-interface in solution.** To characterize the α/β-interface of the B4.2.3 TCR in solution, we

utilized a TROSY-based cross-saturation transfer experiment[34] in which TCR was prepared with $^{15}$N, $^2$H-labelled β-chain and unlabelled (protonated) α-chain. Here selective irradiation of the aliphatic protons of the α-chain is expected to transfer to β-chain amides located at the interface with the α-chain through cross-relaxation. Peak intensity ratio analysis of β-chain amide resonances from saturated relative to non-saturated control experiments revealed cross-saturation transfer effects occurring from the α-chain to β-chain along discrete surfaces. These include a patch of residues on Vβ near the α/β interface (Gly41, Leu42, Gln43, Cys90, Phe101) and a more extended surface area on Cβ that includes residues near the H3/H4 helix regions (Ala137, Val140, Arg187, Val188; Fig. 4a,b). Importantly, β-chain residues distal from the α/β interface are not affected by α-chain saturation as expected (Fig. 4a,b). Mapping of affected β-chain residues on the X-ray structure of free B4.2.3 TCR suggests that the α/β-interface in solution is in good agreement with the crystallographic interface, and is dominated primarily by contacts between the constant domains (Fig. 4b). These NMR data are consistent with analysis of the X-ray structures of the TCR, which reveal that the Cα/Cβ interface has nearly twice the surface area as that of Vα/Vβ (1216 versus 698 Å²). Examination of a number of different TCR X-ray structures indicates that this is a generally observed phenomenon.

**NMR reveals pMHC-binding effects on the TCR β-chain domain.** The completeness of the backbone assignments of unliganded TCR β-chain (Fig. 3a) allowed us to probe local conformational

changes upon binding to its p/MHC ligand. The complex between the TCR ectodomains and P18-I10/H2-D$^d$ is 94.7 kDa and is therefore difficult to characterize by standard NMR methods. Here the use of extensive deuteration of side-chain protons to improve $^{13}$C relaxation and the application of TROSY methods at high magnetic fields to improve $^{15}$N relaxation allowed us to obtain high-quality NMR spectra of the bound state at room temperature (Fig. 3b). In agreement with the previous SPR and AUC measurements (Fig. 1b,c), B4.2.3 TCR formed a tight complex with P18-I10/H-2D$^d$ under the NMR sample conditions. The exchange between free and bound states of the TCR was slow on the chemical shift timescale, as indicated by a single set of peaks for the complex, with large chemical shift changes relative to the free state (up to 0.5 p.p.m. when scaled relative to the $^1$H field, as shown along the β-chain sequence in Fig. 5c). Analysis of the changes in peak positions and intensities revealed two types of effects on the TCR β-chain amide resonances: (1) chemical shift perturbations (Fig. 5c, black bars), indicating a change in the local magnetic environment and (2) conformational exchange-induced line-broadening in the bound state (Fig. 5c, red bars), suggesting a perturbation in μs–ms timescale dynamics or increased solvent exchange rates. Several of the observed chemical shift changes correlate with the displacement of residues in Vβ domain loops seen in the X-ray structures, such as Ser27 in CDR1β, Asp52 in CDR2β and several residues in CDR3β (Fig. 5a, right panel). In addition, the peak intensities of residues 49–53 located in the CDR2β loop were significantly attenuated in the bound form (Fig. 5c). Notably, residues 100–105 located at

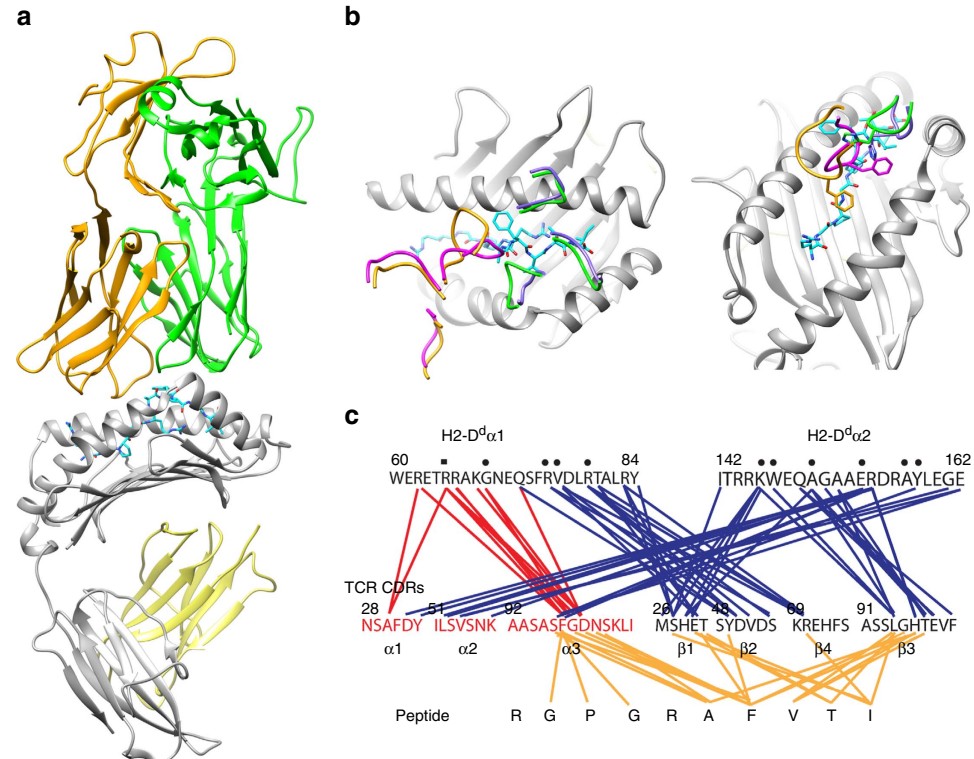

**Figure 2 | X-ray structure of the complex between TCR and pMHC.** (**a**) Ribbon diagram shows the B4.2.3 αβ TCR (above) focusing its CDR loops on the P18-I10/H2-D$^d$/β2m complex (below). TCR α-chain is gold, β-chain is green, peptide (shown in stick representation) is cyan, MHC heavy chain is grey and β2m light chain is yellow. (**b**) enlargement of view of MHC-binding groove and bound peptide. MHC and peptide coloured as in **a**, CDRs of unliganded TCR α-chain are shown in gold, β-chain in green; of liganded TCR α-chains are magenta and of β-chains are purple. The right hand panel (an ∼45° counterclockwise rotation of the left) highlights the large movement of CDR3α residue Phe97. The crossing angle of the TCR on the pMHC, calculated according to Rudolph et al.[19], is 30°. (**c**) Contact map illustrates interactions between TCR and peptide residues (in gold), and between TCR and MHC helices (in red and blue). Data collection and refinement statistics are provided in Table 1. Structural changes between the free and bound forms of the TCR are outlined in Supplementary Fig. 1.

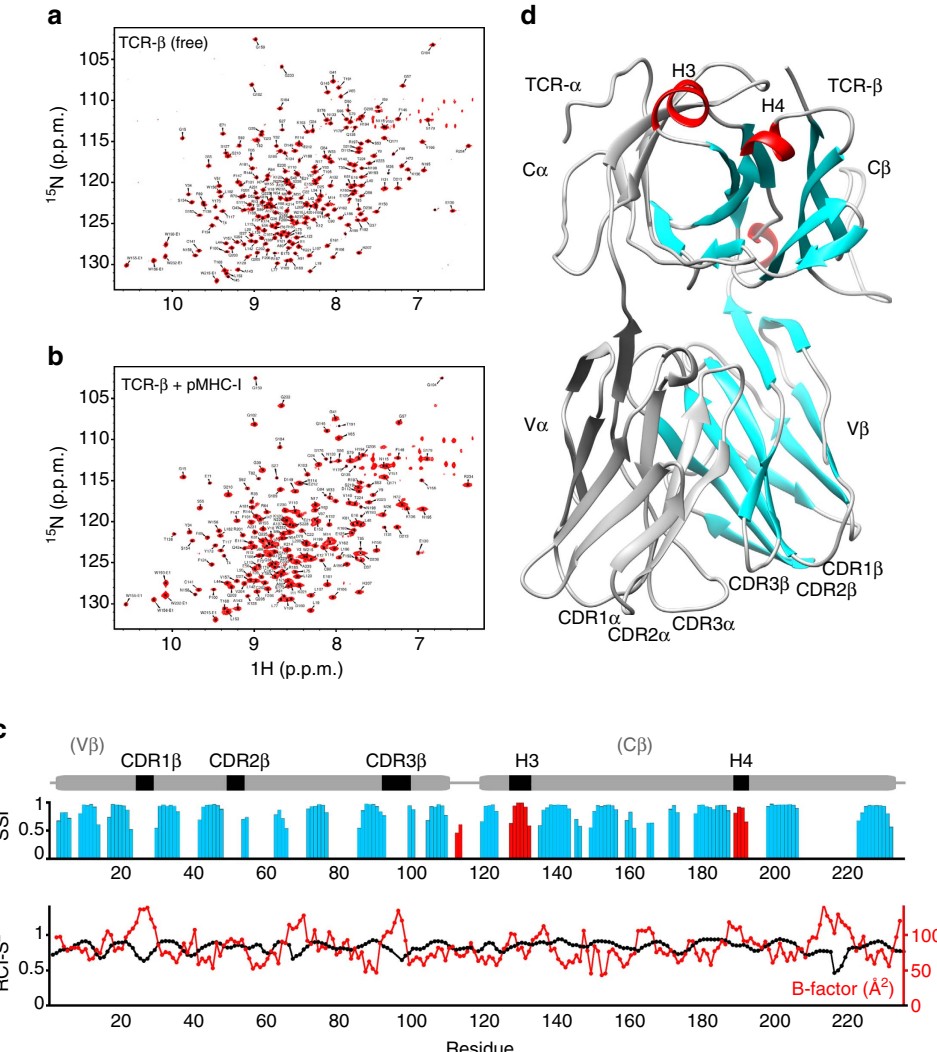

**Figure 3 | NMR characterization of a TCR and its pMHC complex.** (**a**) amide 2D $^1$H-$^{15}$N TROSY-HSQC (900 MHz $^1$H field strength) of a 47.6 kDa disulfide-linked B4.2.3 TCR-αβ heterodimer that is $^{15}$N,$^{13}$C,$^2$H-labelled at the β-chain with 100% back exchanged protons at the amide positions during *in vitro* refolding. The α-chain of the molecule is unlabelled. (**b**) Spectra of the 94.7 kDa 1:1 high-affinity complex of B4.2.3 TCR with P18-I10/H2-D$^d$/β2m ($K_D \sim 0.5$ μM). To improve signal-to-noise for the larger complex, a 16-fold increased number of scans was used when acquiring the data. (**c**) TALOS-N secondary structure index (SSI) derived from the combined $^1$H, $^{15}$N and $^{13}$C chemical shifts along the TCR β-chain sequence. Blue bars of increasing height indicate regions with high β-sheet propensity based on chemical shifts, while red bars correspond to regions with a strong α-helical signature. Gaps correspond to predicted loop regions. The random coil index order parameter (RCI-S$^2$) is also shown, with values <0.75 further indicating unstructured coil regions in solution. As a comparison, the crystallographic B-factors of amide N atoms in the X-ray structure of the free TCR β-chain are also shown in red. The domain diagram of the β-chain is shown on the top of the plot as guide. (**d**) X-ray structure of the disulfide-linked TCR-αβ heterodimer in the free form, showing the unlabelled α-chain (grey) and labelled β-chain, highlighting the CDRs in the Vα, Vβ domains and short helical regions in the Cβ domain (H3, H4). The TALOS-N-derived secondary structure based on the NMR chemical shifts is also shown on the structure with same colours as in **c** and grey for loops, with cartoons drawn according to the DSSP annotation of the PDB entry. CDR, complementarity-determining regions. H3, H4 Cβ domain α-helices 3 and 4.

the Vα/Vβ domain interface also showed above average chemical shift perturbations (Fig. 5c), indicating that p/MHC-binding-induced conformational changes in the V domains are not restricted to the CDR loops.

Strikingly, the NMR data revealed long-range effects on the Cβ of the TCR, at the membrane proximal face of the molecule. These changes are highly localized, and cluster near the H3 helix of Cβ (Glu130, Thr138) at the interface with the Cα domain (Fig. 5c). In addition to the shifted peaks, the resonances of several amides in the region were significantly attenuated in the bound form, suggesting further changes in dynamics at or near the H3 and H4 regions. These include Ser127, Lys134, Ser183,

Arg187 and Val188, all located at the interface with Cα (highlighted with red asterisks in Fig. 5c and shown as red bars in Fig. 5a, left panel). Since amide chemical shifts can be influenced by the local backbone conformation and hydrogen-bonding geometry, these results point to discrete structural changes in the H3 and H4 regions of the β-chain of the B4.2.3 TCR upon p/MHC binding. The structural changes observed in solution in the region of the H3 helix prompted us to compare available liganded and unliganded X-ray structures of MHC-I-restricted TCR[19]. Although there are a limited number of X-ray structures of pMHC-I/TCR complexes for which comparison with the unliganded TCR may be made, careful

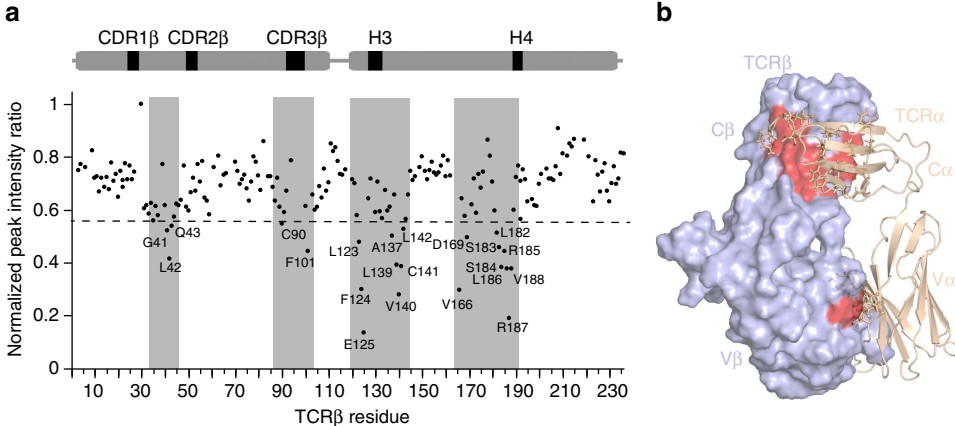

**Figure 4 | Cross-saturation transfer NMR elucidates intermolecular contacts of the αβ-TCR in solution.** (**a**) Cross-saturation transfer experiment performed on free U-[$^{15}$N, $^2$H] β-chain-labelled and unlabelled (protonated) α-chain B4.2.3 TCR acquired at 800 MHz, 25 °C. Experimental details are outlined in Methods. The plot shows the normalized peak intensity ratio ($I_{saturated}/I_{non-saturated}$) as a function of TCRβ residue. The dashed line signifies 1 s.d. of chemical shift change from the average. (**b**) Mapping of TCRβ residues affected upon cross-saturation transfer (shown in red) on the X-ray structure of the B4.2.3 TCR with β-chain shown as a surface display (light blue) and α-chain shown as cartoon (wheat) with side chains of α-chain residues near the α/β interface shown as sticks.

inspection of 10 pairs of superposed structures in the H3 helix revealed no differences in backbone configuration and no consistent changes in side-chain orientations. Notably, the H3 helix does not participate in any crystallographic contacts in our study.

We additionally carried out NMR titrations of B4.2.3 TCR selectively methyl-labelled in its β-chain with p/MHC to complement our amide-mapping results. Methyl groups, such as Ala Cβ, Ile Cδ1, Leu Cδ1/Cδ2 and Val Cγ1/Cγ2, are highly sensitive probes of side-chain packing within hydrophobic cores of proteins and are less influenced by the molecular size of the system under investigation because of their favourable relaxation properties[35]. The quality of the 2D $^1$H-$^{13}$C HMQC spectrum of fully assigned AILV-methyl labelled β-chain TCR, refolded with unlabelled α-chain (Fig. 6a, red), allowed us to probe dynamic changes that occur in the P18-I10/H2-D$^d$-bound state (Fig. 6b, blue). To transfer the methyl assignments to the bound state, we recorded a 3D $H_M$-$C_M H_M$ SOFAST NOESY data set on the P18-I10/H2-D$^d$/B4.2.3 complex and compared it to a similar experiment recorded for the free state (Supplementary Fig. 4e). Similar to the $^{15}$N-TROSY results obtained for β-chain backbone amides (Fig. 5c), we measured highly reproducible slow-timescale changes in peak positions (Fig. 6a, arrows) and intensities (Fig. 6a, asterisks). In particular, we observed significant effects for the methyls of Leu42, Ile45 and Ile47, located on the β-strand of the Vβ domain leading to CDR2β, as well as Val51 located on CDR2β (Fig. 6b, top panel). This corroborates the changes observed in the X-ray structure of the p/MHC-bound TCR relative to the apo structure (Supplementary Fig. 1). Smaller but statistically significant changes were also observed for the resonances of Leu75, Leu77 and Val87 located near the core of the Vβ domain (Fig. 6b, top panel). Likewise, the change in peak intensity between the free and bound states revealed effects: (1) near and on the CDRs including Val3, Ile47, Val51, Leu94, Val99; (2) near the interface between the Vβ and Cβ domains including Ala80, Val116; and (3) in the H3 and H4 helix regions including Ala132 and Ala190 (Fig. 6b, bottom panel, Fig. 6c).

**Mutagenesis suggests a role for the TCR Cβ H3 helix.** Our observation that the resonances of residues in and near the Cβ H3 helix of the B4.2.3 TCR undergo significant changes in the bound

state led us to examine the functional role of this putative allosteric site in cell-surface expression and signalling. We targeted Cβ residues that demonstrate significant NMR effects, that is, Ser127, Glu130, Asn133, Lys134 and Thr138 (Fig. 5a,c). We examined single Ala substitution mutants of each of these residues for their effects on cell-surface expression when paired with the parental α-chain. As shown in Fig. 7a, Ala mutation of Glu130 or Thr138 abolishes cell-surface expression as shown by anti-Vα2 antibody staining that was indistinguishable from untransfected controls. Notably, Ala mutations at the same residues in soluble β-chain constructs result in inability to refold into functional TCR heterodimers *in vitro*, consistent with their placement in a critical region at the interface with the α-subunit of the TCR. The remaining three mutants show levels of surface expression identical to the parental receptor and were further analysed for p/MHC binding and TCR activation in stimulation assays. All three mutant transfectants show decreased interleukin (IL)-2 production when compared to the wild-type transfected cells with the Asn133 Ala mutant showing the greatest reduction (Fig. 7b). Comparison of the amount of IL-2 elicited by 1 µM peptide reveals a reduction from 587 ± 45 pg in the parental transfectant to only 8 ± 4 pg in the Asn133A mutant, corresponding to a 98% decrease in IL-2 levels. Similarly, at a 1 µM peptide dose, the Lys134A mutant shows a decrease of 84% from the wild-type IL-2 levels, while the Ser127A mutation exerts a less severe effect on IL-2 levels, revealing a reduction of only 34% compared to the wild type. The severely attenuated signalling by the Asn133A and Lys134A mutants is not caused by defects in antigen recognition as all three mutants bind P18-I10/H2-D$^d$ tetramers with equal avidity and show the same rate of dissociation as the parental transfectant (Fig. 7c). These results emphasize the functional importance of TCR Cβ residues Asn133 and Lys134 in p/MHC-dependent signalling and highlight the role of Glu130 and Thr138 for correct assembly, association with the CD3 co-receptor and subsequent surface expression.

To examine further whether the Asn133A substitution affected the affinity of the mutant TCR for P18-I10/H2-D$^d$, or the stability of the TCR heterodimer itself, we prepared recombinant Asn133A TCR and compared it with the parental B4.2.3 TCR for binding to immobilized p/MHC in an SPR assay (Supplementary Fig. 7 and Supplementary Table 1). The thermal

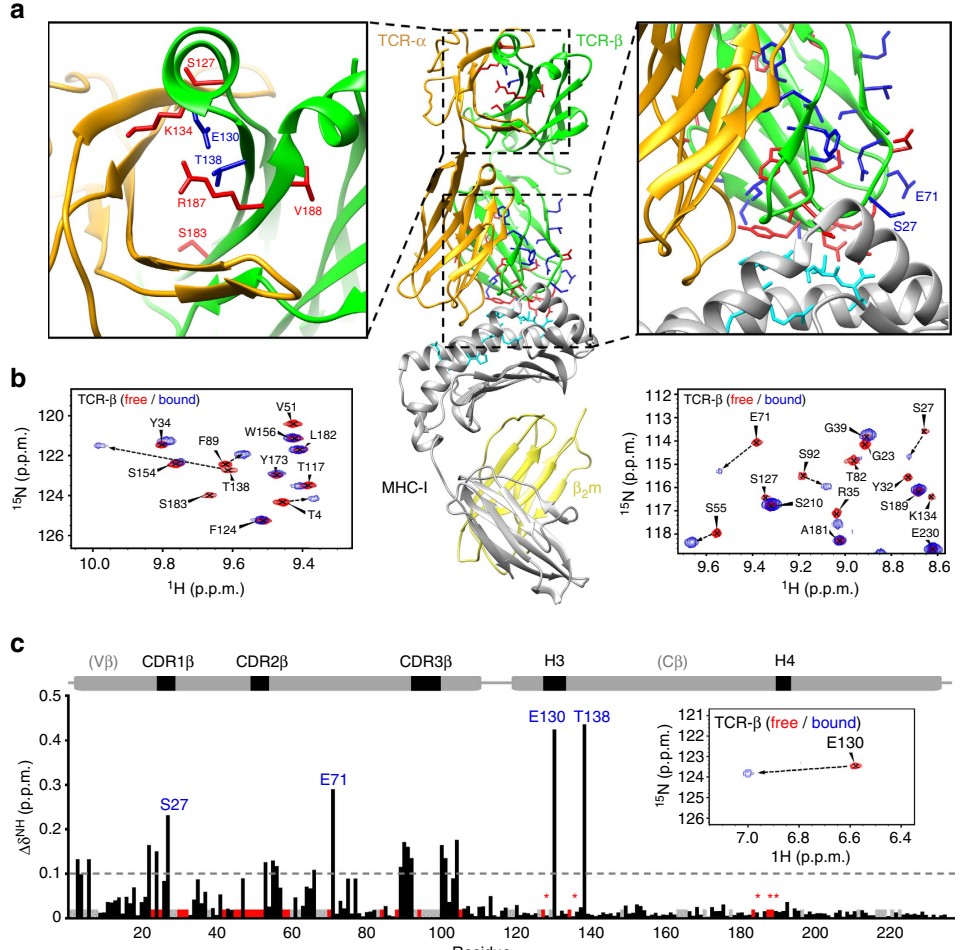

**Figure 5 | pMHC-induced conformational changes on the TCR β-chain domains.** (**a**) TCR β-chain residues with significant amide $^{15}$N-$^{1}$H TROSY-HSQC chemical shift changes upon MHC-I binding are shown as sticks on the Cβ and Vβ domains of the αβ-TCR structure. Two types of observed changes are highlighted with different colours: residues with significantly attenuated resonances (>1 s.d. from the average) in the bound form are shown in red, while residues with significant chemical shift changes are in blue. In the middle panel, the tertiary structure of the B4.2.3 (beige/green)/H2-D$^{d}$ (grey) complex is shown with β2m (yellow) and p18-I10-bound peptide (cyan sticks). (**b**) Overlays of representative regions from TROSY-HSQC spectra recorded in the free (red) and 1:1 MHC-bound (blue) state of the TCR, using two samples that were $^{15}$N, $^{2}$H, $^{13}$C-labelled at the β-chain and unlabelled α-chain (as shown in Fig. 3a,b). Pairs of peaks that correspond to the same residues in the free and bound forms are connected by dotted arrows. (**c**) Summary of combined $^{15}$N/$^{1}$H chemical shift changes along the β-chain sequence, scaled relative to $^{1}$H. The dashed line signifies 1 s.d. of combined chemical shift change from the average. Residues with significantly attenuated resonances in the bound form are shown as red bars, while unassigned and Pro residues are in grey. Red stars highlight Cβ domain resonances that disappear upon p/MHC binding, and that are also illustrated in the structure diagram above (**a**). The large chemical shift change observed for Glu130 upon p/MHC binding is highlighted in the inset.

stability of the mutated TCR was assessed using differential scanning fluorimetry, and was found to be identical to the parental protein (Supplementary Fig. 8). The SPR-binding curves, analysed both kinetically and at steady state, revealed little or no difference in the kinetic association or dissociation rate constants ($k_a$ and $k_d$ values) as well as the calculated or steady-state determined equilibrium constants ($K_D$ values), indicating that the functional effect of the mutation is not the result of a p/MHC-binding defect, but rather reflects the putative allosteric site.

### Discussion

A fundamental question in T-cell immunity is the mechanism by which p/MHC engagement by the TCR is relayed to the associated CD3 subunits to initiate intracellular signalling. Signal transduction likely occurs through ligand-induced conformational changes in the TCR constant domains, which are then

transmitted to the CD3 subunits[15,36–38]. Such structural changes have been difficult to visualize crystallographically, because of the relatively low resolution typically seen for these regions in crystal structures. NMR offers an alternative approach to identify subtle changes in the local magnetic environment of proteins and can thus pinpoint sites undergoing conformational exchange in the NMR timescale. Several studies have used NMR to study the mechanism of p/MHC recognition by the TCR[39,40], or to explore the binding footprint of the CD3 co-receptor subunits. Early NMR studies found the interaction of CD3δε with TCR in solution too weak to detect significant chemical shift perturbations[41]. More recent work found that only a mixture of CD3γε and CD3δε subunits produced detectable effects on the H3 and H4 helices of the TCR Cβ[25], while a subsequent study reported weak but measurable perturbations in both Cα and Cβ upon addition of CD3δε or CD3γε, respectively[26]. On the cytoplasmic side, NMR studies have revealed lipid-sensitive

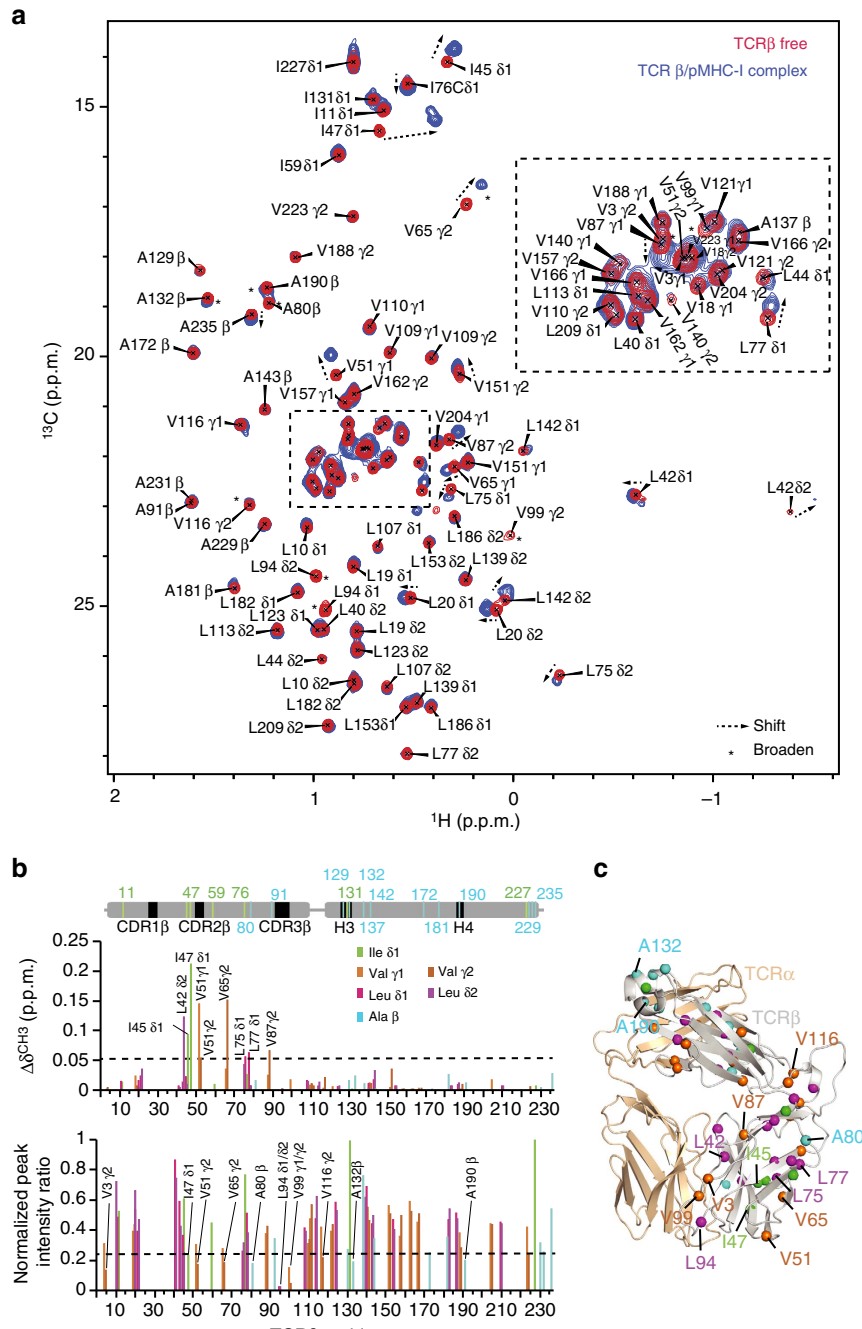

**Figure 6 | Methyl probes of TCR and pMHC complex formation.** (**a**) 2D $^1$H-$^{13}$C HMQC spectra of AILV $^{13}$C-methyl labelled U-[$^{15}$N, $^2$H] β-chain B4.2.3 TCR in the free (red) and 1:1 P18-I10/H2-D$^d$ bound (blue) states acquired at 800 MHz, 25 °C. Changes in chemical shift position are indicated with arrows, while changes in chemical shift intensity due to line broadening are indicated by asterisks. Stereo-specific chemical shift assignments are indicated. Assignment of AILV-methyl chemical shifts is outlined in Supplementary Figs 2–5. (**b**) Plot of chemical shift deviation (CSD in p.p.m.; top) and normalized peak intensity ratio ($I_{bound}/I_{free}$; bottom) as a function of TCRβ methyl residue number. The dashed line signifies 1 s.d. of combined chemical shift change from the average. Affected residues are indicated. (**c**) Mapping of Ala, Ile, Leu and Val methyl probes on the X-ray structure of B4.2.3 TCR with α-chain coloured wheat and β-chain coloured grey. Residues that are affected in either CSD or intensity ratio between the free and bound states are indicated.

conformational changes in the signalling domains of the CD3ζζ homodimer[42] and the CD3ε subunit[38].

Here we use NMR to elucidate changes in TCR dynamics upon binding to the p/MHC ligand. Using a complementary combination of backbone amide and side-chain methyl probes, we extensively map ligand-induced chemical shift perturbations along the TCR β-chain. Consistent with our X-ray structures of the free and p/MHC-bound forms of the receptor, the Vβ CDR

loops that directly engage the p/MHC ligand show large changes in our NMR spectra. In particular, we report significant chemical shift perturbations for the amide resonances of Ser27 on CDR1β, Asp52 on CDR2β and several CDR3β residues, as well as the methyl resonances of Val51 on CDR2β and Leu94, Val99 on the hypervariable loop region. Notably, we also observe significant NMR effects on sites located near the H3 and H4 helices of the Cβ domain, distal to the p/MHC recognition site. These include

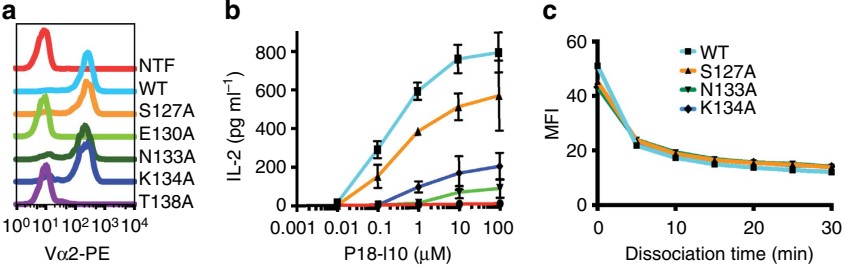

**Figure 7 | Expression and function of parental and Cβ-mutant TCRs in transfectants of αβ⁻ cells.** (a) Staining of nontransfected (NTF), wild type (WT) and the indicated B4.2.3 TCR Cβ mutants with PE-conjugated anti-Vα2 antibody as analysed by flow cytometry. (b) IL-2 amounts in culture supernatants of nontransfected (NTF), wild type (WT) and the indicated B4.2.3 TCR Cβ mutants stimulated for 16 h with graded concentrations of P18-I10 peptide in the presence of A20 cells. IL-2 levels were quantified by ELISA. (c) Dissociation of P18-I10/H2-D^d tetramers from WT and mutant TCR expressing transfectants detected by flow cytometry at 5-min intervals ranging from 0 to 30 min. Error bars represent mean ± s.d.

the amides of Glu130 and Thr138 and the methyls of Ala132 and Ala190. The importance of the H3 helix for signalling is further corroborated by functional data obtained for selected mutants that inhibit signalling without affecting the expression levels of the TCR, its stability or its ability to recognize the p/MHC.

The potential of TCR self-association having an impact on the observed chemical shift and intensity changes is ruled out by several lines of evidence. First, our AUC data are consistent with a fully monomeric TCR sample. Second, the ¹⁵N and ¹³C linewidths are consistent with a predominantly monomeric form of the heterodimer in solution, and remain very similar over a range of concentrations from 70 to 350 μM. Finally, the NMR spectra show the same slow-exchange process and changes over a range of TCR concentrations from 70 to 250 μM. These results conclusively support the view that the significant distal changes observed by NMR arise from an allosteric communication mechanism, as opposed to a secondary binding site or transient dimerization.

Close inspection of interactions near the H3 region in the crystal structures of the free and p/MHC-bound forms of the TCR suggests a possible explanation for the observed NMR chemical shift perturbations at this site. The interface between the Cα and Cβ domains of the TCR is composed primarily of polar and charged residues that form a network of electrostatic interactions[22]. In particular, Tyr125α in the unliganded TCR interacts with Asn133β and Lys134β in one of the three molecules in the asymmetric unit (Fig. 8a) and has long-range contacts with these residues in the other two molecules. However, a displacement of the Cα domain upon p/MHC-binding promotes the formation of a Glu121α-Lys134β salt-bridge in the bound state (Fig. 8b). Notably, the side chains of Arg187β and Asp142α interact in both the free and the bound structures (Fig. 8a,b). These structural displacements are consistent with the measured chemical shift perturbations for the Glu130β, Thr138β and the broadening of the Lys134β, Arg187β backbone amide resonances (Fig. 5c), as well as the broadening of the Ala132β (on H3), Ala190β (on H4) methyl resonances (Fig. 6b). Thus, solution NMR mapping and X-ray structural data are consistent with a systematic reorganization of the Cα/Cβ interface involving a remodelling of electrostatic interactions near the H3 and H4 regions of Cβ. Taken together, our results provide a link between p/MHC binding at the CDR loops and dynamic changes in the TCR Cβ domain that are critical for signalling[43].

The interface between the Cα and the Cβ domains plays an essential role for the stability and assembly of the TCR heterodimer, as shown both *in vitro* and *in vivo* in a recent study[44]. In accordance, residues at the Cα/Cβ domain interface near the H3 helix are highly conserved across α- and β-chain sequences from different species, which could result from their

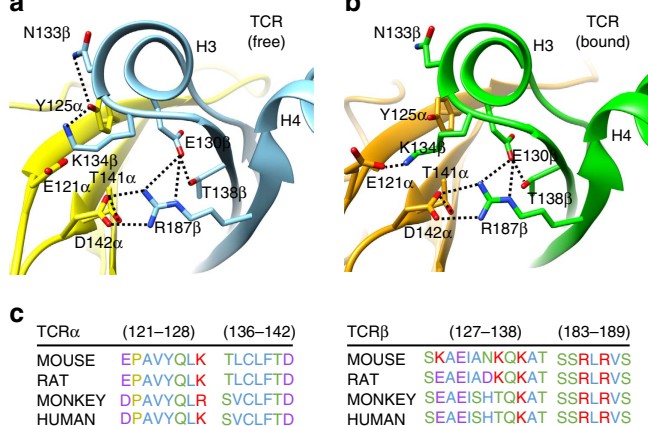

**Figure 8 | Sequence conservation at the Cα- and Cβ-domain interface near the H3 helix of the TCR.** (a) Free and (b) bound X-ray structures of the B4.2.3 αβ TCR dimer showing a network of electrostatic interactions (dashed lines) at the interface between the α- and β-subunit constant domains. The Cα and Cβ domains are shown with yellow/beige and cyan/green in the free/bound structures, respectively. Functionally important residues identified by NMR to show significant changes upon MHC binding are highlighted on the structure. (c) Sequence conservation patterns within the same regions of the Cα and Cβ domains as in **a,b**.

role in stabilizing the TCR itself as well as communicating an activation signal to CD3 (Fig. 8c). While Asn133β and Lys134β are not 100% conserved, residues with potential for hydrogen bond formation are retained at these positions, suggesting that the interface interactions with Tyr125α and Glu121α are crucial for TCR stability and CD3 signalling.

A plausible structural mechanism for the transmission of conformational changes from the CDRs to the Cβ distal sites is suggested by the observation of affected sites in the Vβ/Cβ linker region, in particular Val116β, which shows significant line broadening in our NMR spectra. It is therefore conceivable that dynamically driven p/MHC-induced allosteric changes in the constant regions of the TCR, supported both by NMR data presented here and previous hydrogen/deuterium exchange experiments[45], could potentiate interactions with CD3γε and δε, which, according to recent NMR studies, interact only weakly with the free TCR[25,26].

The role of the α-chain in mediating these structural changes cannot be directly addressed with the β-chain labelling scheme used here. However, similar effects on the α-chain structure are implied by the fact that most sites that undergo conformational

transitions are localized near the α/β-interface. This is consistent with an earlier crystallographic study of a human TCR that revealed extensive conformational rearrangement of the BC and DE turns on Cα (ref. 46) as well as recent fluorescence[47] and hydrogen/deuterium exchange studies[45], which showed that binding of the cognate p/MHC ligand induces long-range effects on the Cα A-B loop region. The structural basis of such changes at the TCR α/β-interface, and their impact on interactions with the CD3 co-receptor should be investigated in future studies.

## Methods

**Protein expression and purification.** The expression and purification of P18-I10/H2-D[d] was performed using standardized protocols[48,49]. Briefly, cDNAs encoding the entire coding sequence of the B4.2.3 TCR α- and β-chains were amplified by RT–PCR from RNA isolated from the B4.2.3 hybridoma[16]. DNAs encoding the extracellular portions of the TCR chains up to, but not including, the membrane-proximal cysteine, were cloned into pET21b (Novagen) and expressed in LB-broth at 37 °C after transformation of *Escherichia coli* BL21(DE3) (Novagen). (All reference to TCR V chain numbering is sequential beginning with the mature TCR chains). Following the strategy described by Boulter *et al.*[50] for human TCRs, an interchain disulfide was engineered between the constant domains of the TCR α- and β-chains by replacing Thr163 of Cα and Ser167 of Cβ with Cys, and the unpaired Cys181 in the Cβ domain was mutated to Ala. TCR-α and -β were expressed separately as inclusion bodies, solubilized in 6 M guanidine-HCl/0.1 mM dithiothreitol (DTT), mixed together and refolded and purified as for P18-I10/H2-D[d].

**Crystallization and structure determination.** Equal volumes of purified B4.2.3 TCR and P18-I10/H2-D[d], each at 7 mg ml[−1], were mixed and allowed to incubate for 30 min at room temperature before crystallization trials. Diffraction quality crystals of the complex were grown in hanging drops at 18 °C in 15% PEG 3350, 0.15 M MgCl₂, cryoprotected in 15% ethylene glycol in 20% PEG 3350, 0.2 M MgCl₂ and frozen in liquid nitrogen. Crystals of free B4.2.3 were grown in hanging drops at 18 °C in 16% PEG 4000, 0.1 M TRIS pH 8.8, 0.2 M MgCl₂, cryoprotected in 15% ethylene glycol in the crystallization solution and frozen in liquid nitrogen. Diffraction data were collected at the National Synchrotron Light Source, Brookhaven, on beamline X29 equipped with an ADSC Q315 detector. Data were indexed and scaled with HKL2000 (ref. 51). Data collection and refinement statistics are reported in Table 1. Molecular replacement solutions were readily found for the TCR and H2-D[d] with Phaser[52] using the murine AHIII 12.2 TCR (PDB 1LP9) and the previously determined structure of H2-D[d] (PDB 1DDH) as search models. For the unliganded B4.2.3 TCR, the best solution was in the P3₁ space group, but analysis of the data by Xtriage indicated pseudo-merohedral twinning with a twin fraction of 0.45, necessitating the application of twin law h, -h-k, -l during refinement in Phenix[53]. This improved R-free to 0.268 along with better electron density. The electron density map of the third heterodimer was less clear compared to the other two suggesting partial occupancy. The occupancy of chains E and F was adjusted to 0.6. Further manual increase in geometrical restraints led to improved refinement. All structure refinements were carried out in Phenix[53] and validated with Molprobity[54]. Structure graphics were prepared with Pymol (Schrodinger, LLC) and Chimera[55].

**Surface plasmon resonance.** BirA-tagged H2-D[d] refolded with either the P18-I10 peptide or with a motif (MTF) peptide (AGPARAAAL) was biotinylated using BirA ligase and biotin (Avidity, LLC) according to the manufacturer's instructions. Approximately 650 resonance units of biotinylated H2-D[d] was captured on a streptavidin sensor chip SA (GE Healthcare) and soluble TCR, either the parental B4.2.3 or with mutated CDR3 residues, at various concentrations, was injected over this surface at 25 °C. The sensorgrams were corrected for binding to control surfaces and fitted to the Langmuir binding equation for a 1:1 interaction model to determine dissociation constants ($K_D$) using the BIAevaluation 3.2 software or EVILFIT[56]. For the binding experiments described in Supplementary Fig. 7 and summarized in Supplementary Table 1, a BIAcore T200 instrument was employed, and data were fit to kinetic and steady-state models using T200 evaluation software 3.0. The same data analysed with EVILFIT gave similar results.

**Analytical ultracentrifugation.** Analytical ultracentrifugation experiments were conducted in an XL-I analytical ultracentrifuge (Beckman Coulter, Indianapolis, IN) with an An50Ti rotor, following standardized protocols[57]. Briefly, protein samples were prepared by dilution of concentrated stocks with the working buffer (25 mM of HEPES, pH 7.2, 150 mM NaCl). Individual protein molecules were characterized in the primary AUC experiments with a concentration series of 1–10 μM. For the binary interactions, equimolar mixtures of B4.2.3TCR + P18-I10/H2-D[d] or B4.2.3TCR + MTF/H2-D[d] were prepared in a concentration series of 0.1–20 μM. The samples were loaded into standard

double-sector charcoal-filled epon centrepieces with 12- or 3-mm path length and sapphire windows. The sedimentation process of the protein molecules was monitored using both Rayleigh interference and ultraviolet absorbance at 280 nm detection at 20 °C and 50,000 r.p.m. The acquired sedimentation velocity data were analysed with SEDFIT using the $c(s)$ sedimentation coefficient distribution approach[58], from which the signal weighted-average sedimentation coefficient ($s_w$) was obtained by integration. To determine the binding affinity, the isotherm of $s_w$ as a function of macromolecular concentrations was fitted with the 1:1 hetero-dimerization model:

$$s_w(c_{A,tot}, c_{B,tot}) = \frac{c_A \varepsilon_A s_A + c_B \varepsilon_B s_B + c_A c_B K_{AB}(\varepsilon_A + \varepsilon_B)s_{AB}}{c_{A,tot}\varepsilon_A + c_{B,tot}\varepsilon_B}$$

where $c_{A,tot}$ and $c_{B,tot}$ denote the total molar concentration for A and B, $c$ indicates the molar concentration of the free component, $s$ denotes the sedimentation coefficient, $\varepsilon$ denotes the extinction coefficient and $K_{AB}$ ($K_D = 1/K_{AB}$) is the equilibrium association-binding constant. In the analysis, $s_A$ and $s_B$ were fixed at the experimentally determined values, while $K_{AB}$ and $s_{AB}$ were subject to optimization through nonlinear regression. The error surface projection analysis was exploited to determine the error intervals of the best-fit $K_D$ values at a 95% confidence level.

**NMR sample preparation and backbone assignments.** U-[¹⁵N,¹³C,²H]-labelled β-chain B4.2.3 TCR samples for NMR were prepared by substituting the TCR β-chain growth medium with M9 minimal media in ²H₂O containing 2 g l[−1] ¹³C, ²H glucose (Sigma #552151) and 1 g l[−1] ¹⁵NH₄Cl. To promote overexpression, we added 1 g l[−1] ²H,¹⁵N,¹³C ISOGRO (Sigma #608297). The purified β-chain inclusion bodies were refolded with unlabelled α-chain. The protein sample-refolding conditions in aqueous buffer enabled complete exchange of the amide ²H with ¹H atoms. All NMR experiments were recorded at a temperature of 25 °C at 600, 700, 800 and 900 MHz cryoprobe-equipped Bruker and Varian spectrometers. To assign the backbone resonances, we used an array of TROSY-readout triple-resonance assignment experiments (HNCO, HN(CA)CO, HNCA and HN(CA)CB), recorded at 600 MHz, supplemented with a 3D NOESY-[¹H,¹⁵N,¹H]-ZQ-TROSY experiment[59], recorded at 800 MHz. We used standard (incrementally sampled), non-constant-time 3D experiments with optimized INEPT transfer delays and shorter acquisition times in the indirect dimensions (30 ms in ¹⁵N, 20 ms in ¹³CO and 10/5 ms in ¹³C_{α,β}) with a mixed-time ¹⁵N evolution period[60] when needed (HNCA and HN(CA)CB). TCR/MHC complexes for NMR were prepared by mixing β-chain-labelled B4.2.3 αβ TCR and P18-I10/H2-D[d] at 1:1 molar ratios, followed by incubation at room temperature for 1 h. Pure TCR/pMHC complexes were isolated by size exclusion chromatography using a Superdex 200 increase 10/300 GL column (GE Healthcare #28-9909-44) with flow rate 0.5 ml min[−1] in 150 mM NaCl, 25 mM Tris pH 8.0. TCR/pMHC complexes prepared for NMR by size exclusion chromatography isolation and by titration were indistinguishable in terms of the measured chemical shift changes. In contrast, control-binding experiments using the non-cognate ligand (H2-D[d] displaying the motif peptide AGPARAAAL) showed no effects on the NMR spectra of the TCR. For NMR-binding experiments U-[¹⁵N,¹³C,²H]-labelled β-chain B4.2.3 TCR was mixed with unlabelled P18-I10/H-2D[d] p/MHC at a 1:1 molar ratio in identical buffer conditions (25 mM HEPES pH 7.3, 50 mM NaCl, 0.01% NaN₃, 1 U Roche protease inhibitor). The final concentration of the complex sample was 250 μM. The free state assignments of the TCR β-chain were transferred to the 94.7 kDa B4.2.3/P18-I10/H-2D[d] complex sample, by closely mapping to the nearest TROSY-HSQC resonances and further NOE-based validation and Ala mutagenesis for selected peaks. All spectra were processed with NMRPipe[61] and analysed with NMRFAM-SPARKY[62]. Chemical shift perturbations (CSP p.p.m.) for β-chain amide resonances in the free versus the bound state were calculated using the equation $\Delta\delta^{NH} = [1/2 \, (\Delta\delta_H^2 + \Delta\delta_N^2/25)]^{1/2}$ (ref. 63).

**Cross-saturation transfer of β-chain-labelled TCR.** To determine the surface of the β-chain affected by the α-chain in solution, 2D cross-saturation transfer experiments[34] were performed on a 180 μM B4.2.3 TCR sample prepared with U-[¹⁵N,²H]-labelled β-chain and unlabelled (protonated) α-chain in 50 mM NaCl, 25 mM HEPES pH 7.3, 0.01% NaN₃, 1 U Roche protease inhibitor in 90% H₂O/10% D₂O. Acquisition parameters for the 2D ¹H-¹⁵N TROSY-HSQC were 256 and 2,048 complex points in the ¹⁵N, ¹H_N dimensions with acquisition times of 45 ms, 71 ms and 16 scans recorded at 800 MHz, 25 °C. Selective saturation of the α-chain aliphatic protons was achieved using a 1,000 point per 15 ms WURST-20 adiabatic pulse[64] with a 5.3 kHz sweepwidth and maximum radio-frequency amplitude γB1(max)/2π of 237 Hz (Q-factor of 1.0). The centre saturation frequency was set to 0.9 p.p.m. To calibrate the adiabatic pulse inversion profile, we recorded a series of 1D ¹H spectra using a WATERGATE[65] water suppression element, such that the aliphatic region, including H_α, was sufficiently saturated while ensuring minimal perturbation of the water and amide resonances. The recycle delay was set to 3.2 s, of which 1.2 s was used for the block of saturation pulses. A control experiment without saturation of the α-chain was obtained using the same parameters with the saturation pulse turned off. The change in peak intensity was determined by calculating the ratio of $I_{saturated}/I_{non-saturated}$ for each assigned β-chain amide resonance and then the ratio was normalized to 1 based on the most-intense NMR peak (T30).

## ILV* and AILV-methyl sample preparation and methyl assignments.

Methyl-labelled samples of the TCR β-chain were prepared according to standardized protocols[28] using selectively labelled precursors in two distinct $^{13}C$-labelling patterns, referred to as ILV* and AILV herein. All isotopes were obtained from ISOTEC Stable Isotope Products (Sigma-Aldrich), and the catalogue numbers are indicated below. ILV*-methyl (Ile $^{13}CH_3$ for δ1 only; Leu $^{13}CH_3/^{12}C^2H_3$; Val $^{13}CH_3/^{12}C^2H_3$) U-[$^{15}N$, $^{13}C$, $^2H$]-labelled β-chain B4.2.3 TCR was prepared in M9 minimal media culture in 1 l $^2H_2O$ supplemented with 3 g $^{13}C$, $^2H$ glucose (Sigma #552151) and 1 g $^{15}NH_4Cl$ (Sigma #299251). The selective labelling of ILV* methyls was achieved by adding 60 mg l$^{-1}$ 2-ketobutyric acid-$^{13}C_4$,3,3-$^2H_2$ (Sigma #607541) for Ile and 120 mg l$^{-1}$ 2-keto-3-(methyl-d$_3$)-butyric acid-1,2,3,4-$^{13}C_4$, 3-$^2H$ (Sigma #637858) for Leu/Val 1 h prior to induction with 1 mM isopropyl-D-thiogalactoside (IPTG). AILV-methyl (Ala $^{13}Cβ$; Ile $^{13}Cδ1$; Leu $^{13}Cδ1/^{13}Cδ2$; Val $^{13}Cγ1/^{13}Cγ2$) U-[$^{15}N$, $^2H$]-labelled β-chain B4.2.3 TCR was prepared in a minimal media culture in 1 l $^2H_2O$, supplemented with 3 g $^{12}C$, $^2H$ glucose (Sigma #552003) and 1 g $^{15}NH_4Cl$. To promote overexpression, 0.2 g l$^{-1}$ $^2H$, $^{15}N$ ISOGRO (Sigma #608300) was added to the minimal media. The selective labelling of AILV side-chain methyls was achieved by adding 60 mg l$^{-1}$ 2-ketobutyric acid-4-$^{13}C$,3,3-$^2H_2$ (Sigma #589276) for Ile and 120 mg l$^{-1}$ 2-keto-(3-methyl-$^{13}C$)-butyric-4-$^{13}C$,3-$^2H$ acid (Sigma #589063) for Leu/Val 1 h prior to induction and 100 mg l$^{-1}$ L-Alanine-3-$^{13}C$, 2-$^2H$ (Sigma #740055) 30 min prior to induction with 1 mM IPTG.

Both ILV* and AILV TCR samples contained 50 mM NaCl, 25 mM HEPES pH 7.3, 0.01% NaN$_3$, 1 U Roche protease inhibitor in 90% H$_2O$/10% D$_2O$. Under these conditions, we obtained complete $^{13}C$, $^1H$ labelling of the desired methyl groups without observing scrambling to other side-chain carbon atoms, that remained $^{12}C$, $^2H$-labelled. Perfect superposition of Ile $^{13}Cδ1$, Leu $^{13}Cδ1/^{13}Cδ2$ and Val $^{13}Cγ1/^{13}Cγ2$ methyl resonances in the HMQC spectra recorded using the AILV and CT-HMQC spectra recorded using the ILV* sample, respectively, confirmed that the TCR structure and monomeric state were identical in the two samples.

To assign Ile, Leu and Val side-chain methyls, a 3D HMCM[CG]CBCA methyl out-and-back experiment[27] was recorded on 225 μM ILV*-labelled β-chain B4.2.3 TCR at 800 MHz, 25 °C. The use of the ILV*-labelling scheme successfully generates a linear spin system needed for these experiments. Acquisition parameters were 80, 80, 1,536 complex points in the $^{13}C_{aliphatic}$, $^{13}C_M$, $^1H_M$ dimensions with corresponding acquisition times of 4, 10 and 69 ms. A relaxation delay of 1.1 s was used with 40 scans/FID. Chemical shifts ($^{13}C_{aliphatic}$) obtained from the 3D HMCM[CG]CBCA were compared with $^{13}C_α$ and $^{13}C_β$ chemical shifts for Ile, Leu and Val residues assigned from 3D HNCA and 3D HN(CA)CB experiments for unambiguous assignment of side-chain methyl resonances in the 2D $^1H$-$^{13}C$ SOFAST HMQC. The 3D HMCM[CG]CBCA also allowed for the assignment of Ile γ1 and Leu γ chemical shifts. Ala $^{13}Cβ$ methyl NMR peaks in the 2D $^1H$-$^{13}C$ SOFAST HMQC were assigned by comparison with Ala $^{13}Cβ$ chemical shifts observed in 3D HN(CA)CB experiments.

AILV side-chain methyl assignments were validated and stereospecifically disambiguated using methyl-to-methyl NOEs obtained from 3D $H_M$-$C_MH_M$ SOFAST NOESY and 3D $C_M$-$C_MH_M$ NOESY experiments[29]. Here the use of the AILV-labelling scheme outlined above allowed the acquisition of well-resolved $^{13}C$ spectra, without the need for constant-time evolution in the indirect $^{13}C$ dimensions. For the 3D $H_M$-$C_MH_M$ SOFAST NOESY experiment, acquisition parameters were 64, 64, 1,280 complex points in the $^1H_M$, $^{13}C_M$, $^1H_M$ dimensions with corresponding acquisition times of 13, 10 and 50 ms with 16 scans/FID. For the 3D $C_M$-$C_MH_M$ SOFAST NOESY acquisition parameters were 128, 64, 1,280 complex points in the $^{13}C_M$, $^{13}C_M$, $^1H_M$ dimensions with corresponding acquisition times of 20, 10 and 50 ms with 16 scans/FID. Backbone amide and side-chain methyl assignments were cross-validated using methyl-to-amide NOEs obtained from 3D $H_N$-$C_MH_M$ SOFAST NOESY and 3D $C_M$-$NH_N$ SOFAST NOESY experiments[29]. For the 3D $H_N$-$C_MH_M$ SOFAST NOESY acquisition parameters were 80, 64, 1,280 complex points in the $^1H_N$, $^{13}C_M$, $^1H_M$ dimensions with corresponding acquisition times of 9, 10 and 50 ms with 16 scans/FID. For the 3D $C_M$-$NH_N$ SOFAST NOESY acquisition parameters were 80, 64, 1,280 complex points in the $^{13}C_M$, $^{15}N$, $^1H_N$ dimensions with corresponding acquisition times of 13, 13 and 50 ms with 16 scans/FID. All 3D SOFAST NOESY experiments were recorded at 800 MHz, 25 °C on 180 to 320 μM AILV-methyl-labelled β-chain B4.2.3 TCR samples using a relaxation delay (d1) of 0.2 s and NOE mixing time (d8) of 0.3 s. A 2D $^1H$-$^{13}C$ SOFAST HMQC[29] was acquired on 250 μM AILV-methyl-labelled β-chain B4.2.3 TCR in the free state and in the bound state in a 200 μM 1:1 molar complex with P18-I10/H2-D$^d$ at 800 MHz, 25 °C. Acquisition parameters were 250 and 1,280 complex points in the $^{13}C_M$, $^1H_M$ dimensions with corresponding acquisition times of 40 ms, 50 ms using a relaxation delay (d1) of 0.2 s with 64 scans/FID in the free state and 128 scans/FID in the bound state. The change in chemical shift (in p.p.m.) between the free and p/MHC bound state of β-chain AILV-methyls was determined using the equation $\Delta\delta^{CH3} = [1/2 \ (\Delta\delta_H^2 + \Delta\delta_C^2/4)]^{1/2}$ (ref. 63). The change in peak intensity was determined by calculating the ratio of $I_{bound}/I_{free}$ for each AILV-methyl chemical shift and then normalizing the ratio to 1 based on the most-intense NMR methyl peak in the bound state (I227 δ1). To confirm the assignments of β-chain AILV-methyl peaks that shifted upon p/MHC binding, an additional 3D $H_M$-$C_MH_M$ SOFAST NOESY was acquired on labelled β-chain B4.2.3 TCR in a 200 μM 1:1 complex with P18-I10/H2-D$^d$ using 64, 44, 1,280 complex points in the $^1H_M$, $^{13}C_M$, $^1H_M$ dimensions with corresponding acquisition times of 10, 5 and 50 ms and 64 scans/FID.

## Mutagenesis and functional assays.

Mutagenesis was performed using the QuikChange Lightning Multi-Site kit (Agilent Cat No. 210515) following the manufacturer's instructions. Sequences of the mutagenic oligonucleotides are listed in Supplementary Table 2. Full-length parental or mutant B4.2.3 TCR α- and β-chain, linked via a 2A sequence[66] was cloned into the pMXs retroviral vector[67] and transfected into the Phoenix-E ecotropic packaging line using X-treme GENE 9 DNA transfection reagent (Roche). Supernatants were collected after 48 h and used to infect logarithmically growing cultures of 58α$^-$β$^-$, a variant of the DO11.10.7 mouse T-cell hybridoma that lacks a functional TCR[68]. Transductants expressing the B4.2.3 TCR or its mutants were stained with phycoerythrin (PE)-conjugated anti-Vα2 (BD Pharmingen cat. no. 553289 used at 1/100 dilution) and magnetically enriched to >90% with anti-PE Microbeads (Miltenyi Biotech). For stimulation assays, $5 \times 10^4$ parental 58α$^-$β$^-$ or transductants expressing parental or Cβ mutant B4.2.3 TCR were stimulated for 16 h with graded concentrations of P18-I10 peptide in the presence of an equal number of the BALB/c-derived B lymphoma line A20 as presenting cells in 96-well flat-bottom plates (Costar). Supernatants were diluted 1:20 for measurements of secreted IL-2 levels by ELISA (BD-Pharmingen) following the manufacturer's instructions. The results of three independent experiments were combined to obtain the means and their s.d.'s. For tetramer binding and dissociation assays, $10^6$ 58α$^-$β$^-$ cells expressing parental or Cβ mutant B4.2.3 TCR were incubated with 200 ng of PE-labelled P18-I10/H2-D$^d$ tetramer in a volume of 100 μl for 1 h on ice, washed with buffer (PBS containing 2% fetal calf serum and 0.1% sodium azide) resuspended in 0.5 ml buffer containing 4 μg anti-H2-D$^d$ mAb 34-5-8S to block rebinding of dissociated tetramer. At various time points during incubation at room temperature, cells were analysed by flow cytometry for residual bound tetramer.

**Data availability.** The refined coordinates and structure factors for the X-ray structures of free B4.2.3 TCR and P18-I10/H2-D$^d$-bound B4.2.3 TCR have been deposited in the Protein Data Bank (www.rcsb.org) with PDB IDs 5IW1 and 5IVX, respectively. NMR assignments for the backbone and side-chain methyl chemical shifts of the β-chain of the B4.2.3 TCR have been deposited into the Biological Magnetic Resonance Data Bank (http://www.bmrb.wisc.edu) under accession number 26977. All other data are available from the corresponding authors upon reasonable request.

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

## Acknowledgements

We thank Drs Paolo Rossi (University of Minnesota) and Vitali Tugarinov (NIH) for help with the implementation of 3D SOFAST NOESY and 3D HMCM[CG]CBCA experiments, respectively. We additionally thank Howard Robinson for data collection at the NSLS, Brookhaven, NY, Manqing Hong for help in refinement, Jugmohit Toor for fluorimetry and Robyn Stanfield for sharing her script for calculating TCR/pMHC crossing angles. This research was supported by the Intramural research programmes of the NIAID, NIBIB and NIDDK, NIH, a K-22 Career development Award to N.G.S. through NIAID (AI2573-01), and by the Office of the Director, NIH, under High End Instrumentation (HIE) Grant S10OD018455, which provides funding for the UCSC 800 MHz NMR spectrometer.

## Author contributions

K.N., A.C.M., A.B., D.H.M. and N.G.S. designed research, interpreted data and wrote the paper. K.N., J.J., R.W. and L.F.B. prepared samples, performed crystallization and X-ray structure determination, and SPR experiments. H.Z. and P.S. performed AUC experiments and analysed and interpreted AUC data. A.C.M., N.G.S., K.N. and J.Y. prepared isotopically labelled samples and performed NMR experiments. A.C.M., N.G.S. and V.K.K. analysed NMR data. K.N. and M.E.T. performed cell-transfectant experiments.

## Additional information

**Competing interests:** The authors declare no competing financial interests.

