## [Peer Review File · Nature Communications]

Reviewers' comments:

Reviewer #1 (Remarks to the Author):

TCRs transfer the interaction with MHC/peptide complexes to intracellular signals. Currently, the mechanism that transfers this information is not well understood. In the presented manuscript, Sgourakis and colleagues combine biophysical methods to study how the interaction of the MHC/peptide complex is transmitted through the TCR b-chain.

Using SPR and ultracentrifugation, the authors showed that their ligand interacts tightly with the TCR complex. Based on crystal structures of the free and ligand bound TCR they identified regions in TCR that undergo conformational changes upon complex formation, that are located mainly in the binding interface. The authors then use state-of-the-art NMR techniques to accurately probe the interaction and find that regions remote from the direct interaction sites are effected by complex formation. These data thus reveal the presence of long range communication between distant sites in the TCR complex. Based on assays, the authors then show that the site remote from the binding site is functionally important.

The manuscript is well written and the data is presented in a clear manner. I suggest publication of the paper with minor revisions in case the following points are addressed:

Fig 2a. Is it possible to report the on- and off-rates based on the SPR experiment.

Fig 2b. It is not clear to me how the orientation of panels 2B relate to each other.

Page 7: "measured at the Ca atom of His95 " should be His96

Page 8: "Resonances from several residues within the β -chain CDRs were absent in the TROSY spectra ". Are those residues in CDR3 only, or are also other resonances not visible in the NMR spectra.

Fig 3b: How were the assignments of the complex obtained? As the complex is in slow exchange one cannot just follow the CSPs (chemical shift perturbations) during a titration. I also am not sure if it would be possible to perform backbone assignment experiments on the very large complex.

Page 10: "conformational exchange-induced line-broadening in the bound state " Could the attenuation of the signals in the bound state also be due to the enhanced relaxation due to the introduced protons from the binding partners? Please comment.

Fig 4: Panels appear in a different order in the text. Maybe consider moving Fig4c above Fig 4a.

Page 11: "AILV-methyl $^{13}\text{C}/^{1}\text{H}$ labeling of the TCR β - chain on $^{12}\text{C}/$ perdeuterated background, also refolded with unlabeled α -chain ". Why was the methyl labeled protein refolded? This seems not required for the NMR measurements as the methyl protons are not labile.

Page 11-12: "We targeted $\text{C}\beta$ residues that demonstrate significant chemical shift perturbations upon p/MHC binding i.e. Ser127, Glu130, Asn133, Lys134, and Thr138 (Fig. 4a, c)." According to Fig 4c Asn133 does not experience and CSPs. This should be clarified, especially as Asn133 is the mutant that displays the largest functional effect. A more solid relationship between CSPs and functional relevance could/ should be obtained by including more mutations in the functional assays.

To further strengthen the allosteric correlation between the helix 3 and CDR regions one could record NMR spectra of a TCR complex that contains point mutation in the H3 region. In case a strong allosteric network is present this would result in small CSPs in the CDR regions.

Figures showing the electron density of the regions shown in Fig S1b should be included. The resolution of the B423-TCRalone crystal structure is not very high and the B-factors of the CDR1 and CDR2 regions are high, indicating that the side chain orientations might be somewhat inaccurate.

Reviewer #2 (Remarks to the Author):

Natarajan and colleagues describe the identification of a putative "allosteric" site or region in an alpha/beta T cell receptor. The study begins with the biochemical interaction of the B4.2.3 TCR and its interaction with a high affinity ligand. The SPR and AUC is performed with rigor. The structures of the free TCR and its complex are determined with good statistics. Conformational changes of the type commonly observed in hypervariable loops are described. Things get very interesting in the latter half of the paper, where NMR chemical shift analysis reveals changes in the chemical environment of amide nitrogens of residues in the Cb domain distal from the environment. This analysis was followed by analysis of methyl side chains using perdeuterated protein, which showed similar effects and possibly suggesting a "pathway" of communication from the binding site to the constant domain. The impact of changes to some of these residues on T cell recognition to is demonstrated using transduced T cells.

Over the past several years there has been significant discussion about T cell receptor signaling mechanisms, without convergence. Conformational changes remain frequently discussed, yet conclusive data has not been shown. Some have suggested subtle and/or dynamic changes that are not easily detectable by structure may contribute as seen in other systems, yet there has not been much data to support this. One reason is that the NMR studies that could shed light on this question are very challenging. Thus the results of this paper are thus significant and the paper is very appropriate for Nature Communications. Nonetheless, there are some areas that need some attention. In particular there are control experiments lacking, the results of which could help illuminate what is happening at the molecular level.

First, the NMR should be repeated in the presence of the noncognate ligand. This will ensure that we are not looking at some form of protein concentration dependent effect. Yes, the AUC data do not show aggregation, etc., but as the authors note NMR is sensitive to subtle shifts that AUC might not detect. As assignments are in hand, this is an easy control that would only need to be performed with the ¹⁵N-labeled sample.

Second, the authors should look further into the impacts of the various mutations they explored in the signaling experiments. It isn't clear whether in vitro protein was produced for these – on p. 13 it is mentioned that "Notably, Ala mutations at the same residues in soluble beta-chain constructs result in the ability to refold...in vitro..." Presumably the others were ok? Data for this should be shown. The authors should verify protein stability; this could be done by looking at T_m values measured by CD or scanning fluorimetry. The results could help us understand what is happening in the protein itself, as surface expression does not have to correlate linearly with protein stability. Also, I was surprised that the authors did not perform SPR binding experiments with the mutations that signaled. This could further inform what is happening. If binding affinity is not changed at all that would be intriguing (but brings in the complication that impacts to signaling are impacted by direct interactions with CD3 chains....what do the authors think of this?). If binding affinity is reduced, the authors could then more closely link distal changes to impacts on TCR binding. The authors do show that there are no obvious changes in functional avidity or tetramer dissociation, but we are talking about changes in the molecule, and solution binding studies are the more appropriate experiments.

Third, why were none of the Ile positions mutated?

Fourth, although the structures here do not show alterations in H3 that correlate with the chemical shifts, do any other structures show subtle changes here? The authors should examine this region in other situations where structures of free and bound TCRs are available. Different crystal environments may have allowed subtle changes to be observed in other structures, or perhaps there are trends discernable only when multiple structures are compared.

Lastly, the potential for dynamic allostery in influencing TCR signaling has been hinted at before using hydrogen/deuterium exchange mass spectrometry. The authors should reference and briefly mention Hawse et al. JI 2012, "Cutting Edge: Evidence for a dynamically driven T cell signaling mechanism" the results of which are consistent with the work here, but do not go into nearly as much detail and are admittedly difficult to interpret beyond showing changes.

Some minor points:

1) RCI order parameters – these are not the kind of "model free" order parameters that we are used to seeing in studies of NMR dynamics. A couple of sentences are needed re: caveats and limitations of these compared to more traditional S^2 order parameters, as in some cases there have been discrepancies.

2) p. 7 – "Notably...nine out of ten hydrogen bonds...are provided by beta chain CDRs." As TCR complex structures have grown, is this unusual? Some context would be helpful.

3) p. 8 – The extensive interface and hydrogen bonds suggest a structural reason for high affinity. What about the (possible) energetic cost of the CDR3 conformational changes?

4) p. 14 – "...due to the relatively low resolution typically seen for these regions in crystal structures." What about the possibility that these are predominantly dynamic in nature and not reflected in static structures?

Reviewer #3 (Remarks to the Author):

Natarajan et al., describe the interaction between an unusually high affinity TCR and its cognate pMHC utilizing SPR, ITC, X-ray crystallography and NMR. The paper describes significant structural rearrangement of CDR loops upon ligation both crystallographically and via solution NMR in a convincing manner. A key observation appears to be evidence of a structural shift in the region distal to the TCR-pMHC interaction interface. NMR reveals large chemical shift perturbations in this distal region, yet large structural changes are largely absent in the crystal structure. There appears to be some biological significance to the region in question as mutagenesis of residues in this region produce defects in either surface expression of the TCR or loss of IL-2 expression in a t cell hybridoma transfection model.

While much of the data are compelling and the effort of characterizing the TCR-pMHC interaction via NMR is considerable, without the conformational change, the paper may not rise to the significance of this journal. The major question to be answered is whether the putative allosteric effect could be due to self association of the beta chain, which could be changed by pMHC bindin. Thus, this questions regarding the conformational change hypothesis must be addressed satisfactorily before this paper could be accepted.

Questions and Criticisms:

1. Since the largest chemical shift perturbations are in the distal site, it is prudent to ask for a supplemental figure showing definitive backbone resonance connectivity for the residues in question for both free and bound forms.

2. The most trivial explanation for the chemical shift changes at the distal site is that the pMHC is interacting directly with this region of the TCR. This site may harbor some promiscuity in binding, which is one explanation for the CD3 binding reported in He 2015 (ref 37) and Natarjan 2016 (ref 38). Can the authors explain how such an interaction is ruled out experimentally via cross saturation or some other more conclusive method? Similarly, this chemical shift change could result from a change in self association leading to an indirect loss of association with addition of ligand. Have the authors titrated the TCR for concentration dependent effects?

3. The paper does not show a mechanism for transfer of information from the CDR loops to the H3 helix region. How does a change occur on opposite sides of the molecule without perturbing the middle. There are minor changes between the crystal structures and no apparent intermediate chemical shift perturbations in the NMR data. If this is not the case, the authors should illustrate this pathway more explicitly as well as an explanation for the relatively large magnitude of chemical shift changes for E130, T138.

4. Similarly, any conformational changes should be highlighted in an overlay of the free and bound TCR structure.

5. Does the H3 helix participate in any crystallographic (lattice or otherwise) contacts? This should be clearly stated in the text.

6. The hydrogen bonding pattern is said to change such that "Tyr125 α in the unliganded TCR forms polar contacts with Asn133 β and Lys134 β in at least one of the three molecules in the asymmetric unit". What is the case in the other two of three units?

7. In figure 6, there is a 10% decrease in time zero MFI for the mutants versus WT. Could the authors comment on this?

Reviewer #4 (Remarks to the Author):

This research article investigates the presence of an allosteric site in the T cell receptor constant domain that may be crucial for signal transmission to the CD3 domains. The authors measure the binding affinity of B4.2.3 TCR with P18-I10/H2-Dd antigen through SPR and analytical ultracentrifugation. Then they solve the crystal structures of the unliganded TCR and liganded TCR and highlight the differences seen in the CDR3 loops. Additionally they analyze the pMHC binding effects on the TCR β -subunit by NMR chemical shift perturbation analysis and identify that H3 helix undergoes allosteric changes upon pMHC interaction. Finally, the functional effect of the H3 helix in T cell signaling is analyzed.

The work adds some new evidence in analyzing the effect of pMHC engagement on molecular changes in the TCR constant domain. However, overall I am concerned with the novelty and significance of many of the experiments performed especially since the importance of H3 helix in T cell signaling has been previously established in toe recently published manuscripts (Natarajan et al, 2016, Cell Reports; He et al, 2015, JBC). Moreover, Natarajan et al., reported concerted structural shifts in CDR residues upon TCR-CD3 $\gamma\epsilon$ interaction indicating allosteric effects. And therefore the proposal of allosteric effects is not novel.

Because of the lack of novelty and the following concerns listed below this work does not im my opinion warrant publication in Nature Communications.

Major concerns:

1) In Results section: 'B4.2.3 TCR binds to P18-I10/H2-Dd ligand with usually high affinity', the authors use SPR and AUC to measure the high affinity between TCR and pMHC. However, the authors do not address how this makes the TCR unique and how it could relate to its biological function.

2) In Results section: 'The TCR β -chain has regions with increased dynamics in solution', the authors have not studied the TCR α -subunit by NMR. This is a major concern because recent work has shown the importance of TCR α -subunit in CD3 interaction (Natarajan et al., Cell Reports, 2016, Krshnan et al., PNAS, 2016, Birnbaum et al., PNAS, 2014). Natarajan et al., have shown that obtaining backbone assignments of the TCR α -subunit is possible. I fail to comprehend why the authors have not made an effort to study TCR α -subunit by NMR. The authors assume that the TCR β -chain is more important in signal transmission than TCR α -subunit. Previously, AB loop of Ca is shown to undergo a large conformational change upon ligand interaction. So, studying TCR α -subunit is extremely important for thoroughness conclusions of the work described in this manuscript.

3) In Results section: 'The TCR β -chain has regions with increased dynamics in solution', the authors detail the TCR dimer preparation by using labeled β -subunit and unlabeled α -subunit. However, previous works studying TCR by NMR has already established this methodology (Natarajan et al., Cell Reports, 2016; He et al., JBC, 2015). The authors need to credit earlier work in the section where these data are described and cite the previous reports appropriately.

4) In Results section: 'p/MHC binding effects on the TCR domain structures revealed by NMR', the authors perform a direct 1:1 p/MHC addition instead of a titration with lesser amounts of p/MHC. This is important to rule out any artifacts in transferring assignments between unliganded and liganded states. In figure 4b, 4c, intermediate points in the spectra should be indicated if a titration was performed especially since the peak shifts are huge.

5) In Results section: 'p/MHC binding effects on the TCR domain structures revealed by NMR', the authors use AILV-methyl labeling of the TCR β -chain to study p/MHC interaction. However, I wonder what additional useful information these experiments provide which cannot be obtained from ^{15}N -TROSY chemical shift perturbation studies with p/MHC. In Figure 5d, is the I131 shift observed significant? The authors must provide the CSP vs residue number plot for individual amino acid labels similar to Figure 4c.

6) In Results section: 'Mutagenesis and functional data suggest a role for the H3 helix of TCR C β in T cell signaling', the authors express H3 helix mutated TCRs and identify their activation abilities. However, the same exact experiment was reported in Natarajan et al., Cell Reports, 2016 and very similar results were observed. So, I fail to understand what value this entire section adds to the already reported results?

Minor concerns:

1) In Introduction section, the authors need to cite references in the opening paragraphs about recombinatorial assortment, thymic selection, differentiation, etc.

2) In page 4, Reference 15 is about CD3 ϵ and not about CD3 ζ as stated.

3) In Results section, 'TCR uses conformational plasticity within the CDR3 loops to recognize p/MHC', the crossing angle should be indicated in the figure 2b.

4) In page 7, 2nd paragraph, cite earlier original work as well as work reviewing the literature showing large conformational changes seen in CDR loops upon p/MHC interaction (Reiser et al. Immunity, 2002; Rudolph, et al., Ann. Rev. Immunol, 2006)

4) In page 7, Line 18, His96 has been mentioned as His95.

5) In Results section: 'The TCR β -chain has regions with increased dynamics in solution', a list of missing resonances must be provided to better understand the completeness of the data.

6) In page 9, line 9, short 310 helical segment is located in the linker between V β and C β domains and not in 'between C α and C β domains'.

Reviewers' comments & responses:

Reviewer #1 (Remarks to the Author):

TCRs transfer the interaction with MHC/peptide complexes to intracellular signals. Currently, the mechanism that transfers this information is not well understood. In the presented manuscript, Sgourakis and colleagues combine biophysical methods to study how the interaction of the MHC/peptide complex is transmitted through the TCR b-chain.

Using SPR and ultracentrifugation, the authors showed that their ligand interacts tightly with the TCR complex. Based on crystal structures of the free and ligand bound TCR they identified regions in TCR that undergo conformational changes upon complex formation, that are located mainly in the binding interface. The authors then use state-of-the-art NMR techniques to accurately probe the interaction and find that regions remote from the direct interaction sites are effected by complex formation. These data thus reveal the presence of long range communication between distant sites in the TCR complex. Based on assays, the authors then show that the site remote from the binding site is functionally important.

The manuscript is well written and the data is presented in a clear manner. I suggest publication of the paper with minor revisions in case the following points are addressed:

We thank the reviewer for her/his kind remarks on our work. Below are point-by-point responses to comments:

Fig 2a. Is it possible to report the on- and off-rates based on the SPR experiment.

We thank the reviewer for asking this question. On and off rates of the binding of the TCR and mutant N133A have been determined in the characterization of the mutant binding, now shown in Extended Data Figure 7 and summarized in Supplementary Table 2, and in p 17 of the revised manuscript.

To us it is remarkable that our initial measurements of the KD (performed several years ago with completely independent preparations of TCR and P18-I10/H2-Dd and examined on a different instrument, a BIAcore 2000) are so close to the measurements performed currently on the T200 instrument. The current data also permitted determination of the kinetics constants summarized in the Table.

Fig 2b. It is not clear to me how the orientation of panels 2B relate to each other.

The legend to Figure 2 has been modified to indicate the approximate rotation of the figure and its intent to highlight the large conformational change in CDR3 α .

Page 7: "measured at the C α atom of His95 " should be His96

This has been corrected in the text.

Page 8: "Resonances from several residues within the β -chain CDRs were absent in the TROSY spectra ". Are those residues in CDR3 only, or are also other resonances not visible in the NMR spectra.

A new supplemental figure (Extended Data Fig. 6) showing the distribution of the 10% missing non-Pro amide resonances in the 2D ^1H - ^{15}N TROSY spectrum (Fig. 3a) of the β -chain has been added in the revised material. Overall, the missing resonances tend to cluster in the CDRs (CDR3 β and CDR1 β), as well as other dynamic regions of the molecule, such as the CC' loop in the constant region and the FG loop. This information has been noted on p 10 of the revised manuscript. Finally, a full list of our resonance assignments both for the backbone and sidechain methyl atoms has been deposited in the BMRB with accession number 26977.

Fig 3b: How were the assignments of the complex obtained? As the complex is in slow exchange one cannot just follow the CSPs (chemical shift perturbations) during a titration. I also am not sure if it would be possible to perform backbone assignment experiments on the very large complex.

We have prepared four new supplementary figures to highlight our assignment strategy for the free and bound forms of the TCR (Extended Data Fig. 2-5). In particular, the assignments of the complex were transferred from the free form using multiple SOFAST datasets of methyl-methyl, amide-amide and amide-methyl NOEs (Rossi *et al*, *JBNMR*, 2016). This is further emphasized in the updated manuscript on

p 8-9 and 14-15. Finally, a complete list of our resonance assignments both for the backbone and sidechain methyl atoms (reaching 90% completeness for backbone and 100% for methyls) has been deposited in the BMRB with accession number 26977.

Page 10: “conformational exchange-induced line-broadening in the bound state “ Could the attenuation of the signals in the bound state also be due to the enhanced relaxation due to the introduced protons from the binding partners? Please comment.

The reviewer raises a valid concern, that was also brought up in discussions within our group but we concluded that additional protons from the (unlabeled) binding partners should not significantly impact the complex relaxation rates. First, a statistical analysis of protein-protein complexes from the PDB performed in our group indicated that even for amide protons at the interface $\sum(r^{-6} HH_{intra}) \gg \sum(r^{-6} HH_{inter})$, suggesting that the contribution of interface protons in relaxation should be minimal. For example, if there was a significant contribution from additional protons on the interface between protein subunits, we would expect to see enhanced relaxation for sites located near the α/β chain interface, since we used protonated α chain to prepare the TCR heterodimer sample. Such a systematic effect was not observed, suggesting that any additional protons from the pMHC ligand should also have a minor effect on the relaxation rates of the bound state, and these (small) effects should be limited to TCR sites along the pMHC interface. Moreover, the observation of widespread line broadening in regions up to 10s of Å away from the binding site (observed both for the backbone amides and AILV methyls, as outlined in detail in the new Fig. 6a, 6b), points to a more global conformational change in TCR dynamics.

Fig 4: Panels appear in a different order in the text. Maybe consider moving Fig4c above Fig 4a.

We have considered this change but the figure appears a lot more cohesive in its current form. Instead, we have reordered the references to Fig 5a and Fig 5b in the text (note that the old version of Figure 4 is now Figure 5 because of the addition of a new main text figure).

Page 11: “AILV-methyl $^{13}C/^{1}H$ labeling of the TCR β - chain on $^{12}C/$ perdeuterated background, also refolded with unlabeled α -chain “. Why was the methyl labeled protein refolded? This seems not required for the NMR measurements as the methyl protons are not labile.

This is correct. However, *in vitro* refolding from purified inclusion bodies is the only efficient approach to obtain mg quantities of labeled TCR to date. This is due to the complex protein fold of the molecule which contains 5 disulfide bonds, including one inter-chain disulfide (Boulter *et al.*, *Protein Eng.*, 2003).

Therefore, although the β -chain ^{13}C methyl protons are derived from the labeled precursors provided in the minimal growth media, refolding together with unlabeled α chain is still required to prepare the NMR sample of the properly conformed T-cell receptor species.

Page 11-12: “We targeted C β residues that demonstrate significant chemical shift perturbations upon p/MHC binding i.e. Ser127, Glu130, Asn133, Lys134, and Thr138 (Fig. 4a, c).” According to Fig 4c Asn133 does not experience and CSPs. This should be clarified, especially as Asn133 is the mutant that displays the largest functional effect. A more solid relationship between CSPs and functional relevance could/ should be obtained by including more mutations in the functional assays.

Figure 5c (old Fig. 4) shows not only those residues that have large CSPs, but also those that disappear on p/MHC binding (asterisked in red). These include Asn133 and this is in part why Asn133 was included in the panel of H3 mutants tested.

To further strengthen the allosteric correlation between the helix 3 and CDR regions one could record NMR spectra of a TCR complex that contains point mutation in the H3 region. In case a strong allosteric network is present this would result in small CSPs in the CDR regions.

We appreciate the suggestion by the reviewer. However, we believe that the data presented in the paper so far (including the now more complete methyl data obtained during the revision process) adequately support our conclusion that the observed allosteric effects are significant, and also have a significant biological result. We thereby would like to address this topic in future work.

Figures showing the electron density of the regions shown in Fig S1b should be included. The resolution of the B423-TCRalone crystal structure is not very high and the B-factors of the CDR1 and CDR2 regions

are high, indicating that the side chain orientations might be somewhat inaccurate.

Thank you for pointing this out. We have included in Extended data, Figure 1, representation of the density maps of CDR3 of α and β . Indeed, the density of the 3.0 Å unliganded TCR is less clear than that of the pMHC/TCR complex at 2.1 Å, but the backbone and large side chains are still clear.

Reviewer #2 (Remarks to the Author):

Natarajan and colleagues describe the identification of a putative “allosteric” site or region in an alpha/beta T cell receptor. The study begins with the biochemical interaction of the B4.2.3 TCR and its interaction with a high affinity ligand. The SPR and AUC is performed with rigor. The structures of the free TCR and its complex are determined with good statistics. Conformational changes of the type commonly observed in hypervariable loops are described. Things get very interesting in the latter half of the paper, where NMR chemical shift analysis reveals changes in the chemical environment of amide nitrogens of residues in the Cb domain distal from the environment. This analysis was followed by analysis of methyl side chains using perdeuterated protein, which showed similar effects and possibly suggesting a “pathway” of communication from the binding site to the constant domain. The impact of changes to some of these residues on T cell recognition to is demonstrated using transduced T cells.

Over the past several years there has been significant discussion about T cell receptor signaling mechanisms, without convergence. Conformational changes remain frequently discussed, yet conclusive data has not been shown. Some have suggested subtle and/or dynamic changes that are not easily detectable by structure may contribute as seen in other systems, yet there has not been much data to support this. One reason is that the NMR studies that could shed light on this question are very challenging. Thus the results of this paper are thus significant and the paper is very appropriate for Nature Communications. Nonetheless, there are some areas that need some attention. In particular there are control experiments lacking, the results of which could help illuminate what is happening at the molecular level.

We thank the reviewer for her/his comments and feedback; we have carried out additional control experiments in response to the comments, as outlined in detail below:

First, the NMR should be repeated in the presence of the noncognate ligand. This will ensure that we are not looking at some form of protein concentration dependent effect. Yes, the AUC data do not show aggregation, etc., but as the authors note NMR is sensitive to subtle shifts that AUC might not detect. As assignments are in hand, this is an easy control that would only need to be performed with the 15N-labeled sample.

Following the reviewer’s suggestion, we performed the control experiment using a U-[¹⁵N, ²H] labeled β -chain TCR sample and D^d refolded with the non-cognate motif peptide ligand (MTF), at a similar concentration range as the sample we had prepared originally using the cognate P18-I10/H2-D^d ligand. Notably, we did not observe any significant binding effects by NMR, such as chemical shift changes. This is consistent with the AUC and SPR data, and conclusively shows that there is no binding, even at high protein concentrations (100 μ M range). This result has been added to the updated text (p 24-25).

Second, the authors should look further into the impacts of the various mutations they explored in the signaling experiments. It isn’t clear whether in vitro protein was produced for these – on p. 13 it is mentioned that “Notably, Ala mutations at the same residues in soluble beta-chain constructs result in the ability to refold...in vitro...” Presumably the others were ok? Data for this should be shown. The authors should verify protein stability; this could be done by looking at T_m values measured by CD or scanning fluorimetry. The results could help us understand what is happening in the protein itself, as surface expression does not have to correlate linearly with protein stability. Also, I was surprised that the authors did not perform SPR binding experiments with the mutations that signaled. This could further inform what is happening. If binding affinity is not changed at all that would be intriguing (but brings in the complication

that impacts to signaling are impacted by direct interactions with CD3 chains....what do the authors think of this?). If binding affinity is reduced, the authors could then more closely link distal changes to impacts on TCR binding. The authors do show that there are no obvious changes in functional avidity or tetramer dissociation, but we are talking about changes in the molecule, and solution binding studies are the more appropriate experiments.

We should clarify our experiments with the indicated mutants. As shown (Figure 7) we transduced the indicated mutants into an appropriate T cell line and demonstrated that two mutants fail to express at the cell surface (a) and others expressed at levels equal to the parental TCR (a). The tetramer binding studies of the transfectants indicated that those mutants that were expressed were equivalently stable at the cell surface (Figure 7c). In addition, although we were unable to refold several of the mutants successfully for in vitro studies, we were able to prepare Asn133A (the mutant with the greatest functional effect) for binding studies, and now present its binding behavior in comparison to that of the parental TCR in Extended Data Figure 7 and Supplementary Table 2. Clearly this mutation has little or no impact on pMHC binding either as evaluated kinetically or by steady state analysis. Finally, we have tested the impact of the same mutation on the stability of the TCR using scanning fluorimetry, as suggested by the reviewer. The results, shown in Extended Data Fig. 8 are indistinguishable from the WT curve, suggesting that the mutated TCR is properly conformed to bind the p/MHC. These important points have been further highlighted in the revised manuscript, at p 17.

Third, why were none of the Ile positions mutated?

We were particularly interested in mutants that might yield functional differences without affecting expression in transductants, and thus focused on those that we suspected would not affect the TCR core structure. A comprehensive mutagenic analysis was not the goal of our studies.

Fourth, although the structures here do not show alterations in H3 that correlate with the chemical shifts, do any other structures show subtle changes here? The authors should examine this region in other situations where structures of free and bound TCRs are available. Different crystal environments may have allowed subtle changes to be observed in other structures, or perhaps there are trends discernable only when multiple structures are compared.

We have examined superpositions of pMHC/TCR and unliganded TCR structures that are available and have found no consistent differences crystallographically in the H3 helix. We have added text on p 14 summarizing this observation.

Lastly, the potential for dynamic allostery in influencing TCR signaling has been hinted at before using hydrogen/deuterium exchange mass spectrometry. The authors should reference and briefly mention Hawse et al. JI 2012, "Cutting Edge: Evidence for a dynamically driven T cell signaling mechanism" the results of which are consistent with the work here, but do not go into nearly as much detail and are admittedly difficult to interpret beyond showing changes.

We regret the oversight of the Hawse reference. We have added two short segments in discussion and reference to this important early study, and how it relates to our current NMR findings (p 18, 22).

Some minor points:

1) RCI order parameters – these are not the kind of "model free" order parameters that we are used to seeing in studies of NMR dynamics. A couple of sentences are needed re: caveats and limitations of these compared to more traditional S^2 order parameters, as in some cases there have been discrepancies.

A more thorough description of the RCI parameters has been added to the main text (p 10).

2) p. 7 – "Notably...nine out of ten hydrogen bonds...are provided by beta chain CDRs." As TCR complex structures have grown, is this unusual? Some context would be helpful.

We have added a sentence providing the context for this observation on p 7.

3) p. 8 – The extensive interface and hydrogen bonds suggest a structural reason for high affinity. What about the (possible) energetic cost of the CDR3 conformational changes?

We appreciate that the reviewer has pointed out the weakness in this incompletely justified argument. We have modified the statement on p 8.

4) p. 14 – “...due to the relatively low resolution typically seen for these regions in crystal structures.” What about the possibility that these are predominantly dynamic in nature and not reflected in static structures?

We have added a phrase on p 18 to incorporate this possible explanation for the relative difficulty in visualizing these regions crystallographically.

Reviewer #3 (Remarks to the Author):

Natarajan et al., describe the interaction between an unusually high affinity TCR and its cognate pMHC utilizing SPR, ITC, X-ray crystallography and NMR. The paper describes significant structural rearrangement of CDR loops upon ligation both crystallographically and via solution NMR in a convincing manner. A key observation appears to be evidence of a structural shift in the region distal to the TCR-pMHC interaction interface. NMR reveals large chemical shift perturbations in this distal region, yet large structural changes are largely absent in the crystal structure. There appears to be some biological significance to the region in question as mutagenesis of residues in this region produce defects in either surface expression of the TCR or loss of IL-2 expression in a t cell hybridoma transfection model.

While much of the data are compelling and the effort of characterizing the TCR-pMHC interaction via NMR is considerable, without the conformational change, the paper may not rise to the significance of this journal. The major question to be answered is whether the putative allosteric effect could be due to self association of the beta chain, which could be changed by pMHC binding. Thus, this questions regarding the conformational change hypothesis must be addressed satisfactorily before this paper could be accepted.

We thank the reviewer for her/his comments. To strengthen our results with respect to our conformational change hypothesis and exclude the possibility of self-association in the NMR tube, we have recorded additional NMR data, that are presented in 2 new main text and 5 new Supplemental figures. The new data were recorded using 4 new TCR samples, prepared with suitable labeling schemes, as outlined in detail below and in the updated Supplementary Methods section.

Questions and Criticisms:

1. Since the largest chemical shift perturbations are in the distal site, it is prudent to ask for a supplemental figure showing definitive backbone resonance connectivity for the residues in question for both free and bound forms.

We have prepared several new supplementary figures to highlight our assignment strategy for the free and bound forms of the TCR (Extended Data Fig. 2). While the assignments of the free form were obtained using a range of complementary J-correlated experiments, further cross-validated by NOEs, the assignments of the complex were transferred from the free form using multiple 3D SOFAST datasets of methyl-methyl, amide-amide and amide-methyl NOEs (Rossi et al, JBNMR, 2016). This is outlined in detail in the new supporting figures Extended Data Fig. 3, 4 and 5 and further emphasized in the updated manuscript on p 8-9 and p 14-15. To address the distal site assignments, we have prepared a detailed figure, Extended Data Fig. 5, where we show our assignment strategy for residues located on the β -chain H3 helix region.

2. The most trivial explanation for the chemical shift changes at the distal site is that the pMHC is interacting directly with this region of the TCR. This site may harbor some promiscuity in binding, which is one explanation for the CD3 binding reported in He 2015 (ref 37) and Natarajan 2016 (ref 38). Can the

authors explain how such an interaction is ruled out experimentally via cross saturation or some other more conclusive method? Similarly, this chemical shift change could result from a change in self association leading to an indirect loss of association with addition of ligand. Have the authors titrated the TCR for concentration dependent effects?

The reviewer raises a reasonable concern regarding the presence of a true allosteric communication pathway, versus a secondary, low-affinity binding site. Such a site would need to be extremely weak (high micro to milli-molar range K_d), so as to remain undetected by AUC. Notably, such a low-affinity binding site is also not observed in our SPR experiments presented in Fig. 1a, a technique that has a detection limit in that range. Nonetheless, following the reviewer's suggestion, we have implemented the cross-saturation experiments described in (*Takahashi et al, SNMB, 2000*), and tested them for the free form of the TCR, prepared with U- $[^{15}\text{N}, ^2\text{H}]$ labeled β -chain and unlabeled (protonated) α -chain, which is the same labeling scheme as in all other amide chemical shift binding data presented in the manuscript. The results for the free TCR are shown in a new Fig. 4, described on p 11-12 of the revised manuscript, with the saturation transfer taking place from α to β along discrete sites on the V and C regions (including residues near the distal H3 site), in good agreement with our X-ray structure of the free form. This result further suggests that the TCR must be predominantly monomeric in solution. However, using this experiment it is not possible to assess the effects of binding to unlabeled pMHC due to the presence of protonated α chain; since the distal site on H3 participates in intimate contacts with the α chain, any saturation transfer effects could be interpreted as coming from either the pMHC or the α chain itself. Finally, in order to ensure that our complex is 100% monomeric, we have prepared two new complex samples (with various amide and methyl labeling schemes. With these samples, the TCR/pMHC complex was isolated using size-exclusion chromatography to purify the 1:1 stoichiometric complex and to remove any excess of unbound pMHC (described in the updated Methods section on p 24-25). The spectra of the complex sample prepared in this manner show exactly the same effects (*i.e.* changes in chemical shift position and differential line broadening) as previously titration prepared TCR/pMHC complexes, thereby further arguing against the presence of a pMHC weak binding. The new spectra are now displayed in the updated manuscript in the new Fig 6a.

Regarding the potential of change in TCR self-association impacting the observed chemical shift and intensity changes, this can be ruled out by several lines of evidence. First, our AUC data are consistent with a 100% monomeric TCR sample, therefore if transient dimer formation occurred, it would need to take place with a K_d in the low millimolar range. Second, the TCR ^{15}N linewidths and average R2 are consistent with a predominantly monomeric size of the sample, and remain constant over a range of concentrations from 70-350 μM . Finally, we have repeated our binding experiments for a range of TCR concentrations from 70-250 μM , and we still observed the same slow-exchange process, with identical effects in our NMR spectra and in particular with respect to the allosteric changes. Taken together, these results conclusively support our original conclusion, that the significant distal changes observed by NMR arise from an allosteric communication mechanism as opposed to a secondary, weaker binding site or transient dimerization in the NMR tube. This point has been highlighted in a new paragraph in the update manuscript (p 20).

3. The paper does not show a mechanism for transfer of information from the CDR loops to the H3 helix region. How does a change occur on opposite sides of the molecule without perturbing the middle. There are minor changes between the crystal structures and no apparent intermediate chemical shift perturbations in the NMR data. If this is not the case, the authors should illustrate this pathway more explicitly as well as an explanation for the relatively large magnitude of chemical shift changes for E130, T138.

The reviewer makes a valid point regarding the underlying structural basis of our proposed allosteric communication mechanism between the variable and constant domains of the β chain. To address this point, we have carried out the complete assignment of all AILV methyls in the free and bound form (Fig. 6a). Using the new assignments, we now were able to identify sites in the linker region between the variable and constant domains that are also broadened in the bound state, particularly V116. This suggests a mechanism in which conformational changes in the $V\beta$ domain can be propagated to $C\beta$ through a conformational change in the domain orientation. The results are highlighted in the new Fig. 6b and outlined in detail in the updated manuscript on p 20. Here, we should also note several accepted models for allostery that do not rely on "direct transfer of information" (see for instance, Tsai, del Sol, and

Nussinov, 2008, JMB; McLeish, Trends in Biochemical Sciences, 2015; and Wagner Chemical Reviews, 2016 116:6370).

4. Similarly, any conformational changes should be highlighted in an overlay of the free and bound TCR structure.

This is provided in the new Extended Data Fig. 1

5. Does the H3 helix participate in any crystallographic (lattice or otherwise) contacts? This should be clearly stated in the text.

We do not make reference to possible H3 helix changes observed crystallographically, and the H3 helix is not involved in crystal contacts in either the liganded or the unliganded structures. This point has been highlighted in the revised manuscript (p 14).

6. The hydrogen bonding pattern is said to change such that "Tyr125 α in the unliganded TCR forms polar contacts with Asn133 β and Lys134 β in at least one of the three molecules in the asymmetric unit". What is the case in the other two of three units?

We have clarified this statement on p 21.

7. In figure 6, there is a 10% decrease in time zero MFI for the mutants versus WT. Could the authors comment on this?

The data plotted are the Mean as tabulated by the flow cytometer. The histograms represent the usual broad range of fluorescence intensities and the time course is indicative of the stability of the interaction of the cell-expressed TCR with the recombinant p/MHC tetramer. Detailed quantitative analysis of the binding of the Asn133A TCR as compared with the parental B4.2.3 TCR was carried out in the SPR experiments detailed in Extended Data Figure 7 and Supplementary Table 2, which establish quantitatively that the Asn133A mutation has no quantitative influence on the kinetics or steady state binding parameters.

Reviewer #4 (Remarks to the Author):

This research article investigates the presence of an allosteric site in the T cell receptor constant domain that may be crucial for signal transmission to the CD3 domains. The authors measure the binding affinity of B4.2.3 TCR with P18-I10/H2-Dd antigen through SPR and analytical ultracentrifugation. Then they solve the crystal structures of the unliganded TCR and liganded TCR and highlight the differences seen in the CDR3 loops. Additionally they analyze the pMHC binding effects on the TCR β -subunit by NMR chemical shift perturbation analysis and identify that H3 helix undergoes allosteric changes upon pMHC interaction. Finally, the functional effect of the H3 helix in T cell signaling is analyzed.

The work adds some new evidence in analyzing the effect of pMHC engagement on molecular changes in the TCR constant domain. However, overall I am concerned with the novelty and significance of many of the experiments performed especially since the importance of H3 helix in T cell signaling has been previously established in the recently published manuscripts (Natarajan et al, 2016, Cell Reports; He et al, 2015, JBC). Moreover, Natarajan et al., reported concerted structural shifts in CDR residues upon TCR-CD3 $\gamma\epsilon$ interaction indicating allosteric effects. And therefore the proposal of allosteric effects is not novel.

Because of the lack of novelty and the following concerns listed below this work does not in my opinion warrant publication in Nature Communications.

We thank the reviewer for her/his comments, but respectfully disagree. Our work builds on these earlier studies but takes them a pivotal step further by revealing the connection between the (minor, but statistically significant) CD3-induced chemical shift changes described in (Natarajan et al, 2016, Cell Reports; He et al, 2015, JBC) and the significant pMHC-induced effects, studied here. Moreover, we employ a novel methyl-labeling technology to study this important system and provide the first complete methyl assignments of any TCR system to date (also deposited in the BMRB). These points have been emphasized in the revised manuscript.

Major concerns:

1) In Results section: 'B4.2.3 TCR binds to P18-I10/H2-Dd ligand with usually high affinity', the authors use SPR and AUC to measure the high affinity between TCR and pMHC. However, the authors do not address how this makes the TCR unique and how it could relate to its biological function.

The uniqueness of the system is that it is amenable to detailed multidimensional NMR analysis—that the NMR spectra of appropriately labelled TCR are robust, well resolved, and that the expression system has allowed a variety of state-of-the-art labelling methodologies to be applied. The relationship to biological function is documented in the transfection and IL-2 stimulation assays presented.

2) In Results section: 'The TCR β -chain has regions with increased dynamics in solution', the authors have not studied the TCR α -subunit by NMR. This is a major concern because recent work has shown the importance of TCR α -subunit in CD3 interaction (Natarajan et al., Cell Reports, 2016, Krshnan et al., PNAS, 2016, Birnbaum et al., PNAS, 2014). Natarajan et al., have shown that obtaining backbone assignments of the TCR α -subunit is possible. I fail to comprehend why the authors have not made an effort to study TCR α -subunit by NMR. The authors assume that the TCR β -chain is more important in signal transmission than TCR α -subunit. Previously, AB loop of C α is shown to undergo a large conformational change upon ligand interaction. So, studying TCR α -subunit is extremely important for thoroughness conclusions of the work described in this manuscript.

Understandably, the reviewer is concerned about the lack of results reporting on the dynamics of the α -chain for the same TCR system studied here. First, we would like to emphasize that our work focuses on understanding pMHC binding effects on the β -chain domains, a very important part of the TCR complex that is expressed first during T cell development. This is not to say that the α chain is not as important. Although we respect the reviewer's personal opinion, we believe that the main conclusion of our paper (i.e. that pMHC binding induces important allosteric effects on a distal site important for signaling) is adequately supported by the range of NMR data recorded for the β -chain. In order to clarify the focus of study, we have revised the title of our paper to "*An allosteric site in the T cell receptor β -chain constant domain plays a critical role in T cell signaling*".

3) In Results section: 'The TCR β -chain has regions with increased dynamics in solution', the authors detail the TCR dimer preparation by using labeled β -subunit and unlabeled α -subunit. However, previous works studying TCR by NMR has already established this methodology (Natarajan et al., Cell Reports, 2016; He et al., JBC, 2015). The authors need to credit earlier work in the section where these data are described and cite the previous reports appropriately.

These early, groundbreaking studies (already cited in introduction) have been further credited in our results section, following the reviewer's suggestion (p 8). Furthermore, while the previously deposited in the BMRB assignments covered 60% (Natarajan et al., - BMRB entry 26751) and 87% (He et al. - BMRB entry 26569) of all β chain backbone amide groups respectively and did not include sidechain methyl assignments, here we present a more complete coverage, reaching 90% of backbone non-Pro residues and 100% for AILV methyls. Our complete assignments can be accessed in the BMRB entry with accession number 26977.

4) In Results section: 'p/MHC binding effects on the TCR domain structures revealed by NMR', the authors perform a direct 1:1 p/MHC addition instead of a titration with lesser amounts of p/MHC. This is important to rule out any artifacts in transferring assignments between unliganded and liganded states. In figure 4b, 4c, intermediate points in the spectra should be indicated if a titration was performed especially since the peak shifts are huge.

There appears to be some confusion about our chemical shift mapping results, and we apologize for not being sufficiently clear about this in our original presentation. Since binding of the TCR heterodimer to the pMHC ligand is slow on the chemical shift time scale, it is not possible to obtain intermediate peak positions by titration. A titration experiment was actually performed in our preliminary mapping of the complex, and gave us the expected hallmark of slow exchange, i.e. two peaks per each affected site with a peak position corresponding to the free/bound forms and a signal intensity relative to the population ratio and T2 of each state. In our final results, we have purified a stoichiometric 1:1 complex sample by SEC and use it to record all complex spectra. As outlined in detail in the new figures **Extended Data Fig. 2-5**, we have relied on a network of NOE connectivities to transfer our assignments to the bound form

from the free form assignments, that were established using J-correlated experiments and cross-validated using NOEs. These points have been further clarified in the revised manuscript, p 8-9 and p 14-15 as well as in the updated Supplementary Methods.

5) In Results section: 'p/MHC binding effects on the TCR domain structures revealed by NMR', the authors use ALLV-methyl labeling of the TCR β -chain to study p/MHC interaction. However, I wonder what additional useful information these experiments provide which cannot be obtained from ^{15}N -TROSY chemical shift perturbation studies with p/MHC. In Figure 5d, is the I131 shift observed significant? The authors must provide the CSP vs residue number plot for individual amino acid labels similar to Figure 4c. The reviewer understandably questions the practical utility of our methyl labeling scheme of the TCR β chain that, due to the lack of complete assignments, was not sufficiently evident in the original version of manuscript. As part of the revisions we have carried out *de novo* methyl assignments, which involved preparing several new samples using 2 different methyl labelling schemes and recording 6 new 3D NMR datasets, as outlined in detail in Extended Data Fig. 2 and Supplemental Methods. The new data allowed us to obtain conclusive assignments for all methyls, in both the free and bound states (we also found one inversion in our earlier assignments involving I131 and I11 out of a total of 91 methyls, that we corrected). With the complete assignments in hand, we report the requested CSD and intensity plot vs residue number in the new Fig. 6a and Fig. 6b, respectively. The highly sensitive methyl-TROSY spectra (S/N is roughly 10-fold relative to the amide TROSYs) allowed us to 1) cross-validate our amide results and 2) obtain coverage of sidechain atoms that may participate in important rotameric and packing changes that cannot be appreciated by considering the backbone amides alone. In particular, the methyl spectra corroborate the observed dynamic changes in H3 (as evidenced by the line-broadening of the Ala 132 C β in the bound form), help us also identify changes for Ala 190 C β on H4 upon binding, and finally enable us to identify a pathway for allosteric communication between the variable and constant domains involving concerted changes at sites located in the linker sequence (Val 116). These points have been further emphasized in the revised manuscript on p 14-15, 20.

6) In Results section: 'Mutagenesis and functional data suggest a role for the H3 helix of TCR C β in T cell signaling', the authors express H3 helix mutated TCRs and identify their activation abilities. However, the same exact experiment was reported in Natarajan et al., Cell Reports, 2016 and very similar results were observed. So, I fail to understand what value this entire section adds to the already reported results? Granted, that similar mutagenesis and functional experiment (on a completely different T cell receptor) were reported by Natarajan et al. However, our paper addresses the B4.2.3 TCR, which is an MHC-I-restricted murine TCR, and the mutant experiments address the structural and NMR dynamic data concerning the B4.2.3 TCR.

Minor concerns:

1) In Introduction section, the authors need to cite references in the opening paragraphs about recombinatorial assortment, thymic selection, differentiation, etc.

We apologize for any oversight and have added an appropriate reference on p 3 (Davis and Chien, in Paul, 2013).

2) In page 4, Reference 15 is about CD3 ϵ and not about CD3 ζ as stated.

We have corrected this discrepancy on p 4.

3) In Results section, 'TCR uses conformational plasticity within the CDR3 loops to recognize p/MHC', the crossing angle should be indicated in the figure 2b.

The calculated crossing angle is mentioned in the text (p 7) and is now indicated in the legend to Figure 2b as well.

4) In page 7, 2nd paragraph, cite earlier original work as well as work reviewing the literature showing large conformational changes seen in CDR loops upon p/MHC interaction (Reiser et al. Immunity, 2002; Rudolph, et al., Ann. Rev. Immunol, 2006)

Thank you for recognizing our oversight, appropriate references have been added on p 7.

4) In page 7, Line 18, His96 has been mentioned as His95.

This has been corrected in the updated manuscript.

5) In Results section: 'The TCR β -chain has regions with increased dynamics in solution', a list of missing resonances must be provided to better understand the completeness of the data.

A new figure (Extended Data Fig. 6) showing the distribution of the (10% in total) missing resonances has been added in the revised material.

6) In page 9, line 9, short 310 helical segment is located in the linker between V β and C β domains and not in 'between C α and C β domains'.

This has been corrected in the updated manuscript (current p 10).

REVIEWERS' COMMENTS:

Reviewer #1 (Remarks to the Author):

The authors have addressed all my concerns and I congratulate them with their achievements. Especially their NMR work is of very high quality and shows the power of solution state methods to address biological aspects in large complexes.

Reviewer #2 (Remarks to the Author):

The authors have done an excellent job responding to my concerns. This is a real tour de force. My only remaining comment is about relates to the novelty that dynamic allostery resulting from pMHC binding could influence interactions with CD3 proteins. This hypothesis was put forward earlier in Hawse et al. 2013 which showed distributed changes in hydrogen/deuterium exchange upon TCR binding (a possible dynamic allostery mechanism was the title of the paper). The authors do cite this paper, but do not mention it terms of dynamic allostery. This could be fixed by simply citing it in the sentence at the end of the first paragraph on p. 20 ("It is therefore conceivable that p/MHC-induced allosteric...". But great job all around, this will be an important paper that should generate good discussion about signaling mechanisms.

Reviewer #3 (Remarks to the Author):

The revised manuscript provides helpful controls, new data and new data presentation that reassure the reviewer that every care has been taken that the results are not due to trivial artifacts or assignment errors. The paper has quite a few new controls and it is convincing that there is little or no self association of the TCRs. They make reasonable arguments that secondary binding sites are not present based on calorimetry and SPR, but they fail to provide a definitive cross saturation experiment with perdeuterated alpha, perdeuterated and 15N beta, and protonated pMHC. Rather, they show cross saturation data between protonated alpha and perdeuterated 15N beta to recapitulate the alpha/beta interface and show that alpha does not associate with beta outside of this interface. Technically, the cross saturation experiment is possible if TCRalpha is perdeuterated and otherwise unlabeled, TCRbeta is 15N/perdeuterated and pMHC is protonated but otherwise unlabeled. Perhaps the size of the complex puts this experiment beyond the authors' capabilities. This omission leaves the possibility of a secondary interaction site between TCRbeta and pMHC open and weakens the paper. As pointed out by the authors, NMR is sensitive to changes not seen by other techniques. This might be just such a case.

In summary, while there is as yet no satisfactory explanation for the magnitude of chemical shift changes, the report appears to show a distal change in the TCRbeta component of TCRalpha-beta with addition of pMHC. Given the lack of mechanism for the distal chemical shift changes and little corroborating evidence in existing paired X-ray structures of liganded and unliganded TCRs, the significance of the isolated chemical shift perturbations is still quite uncertain. The new data showing ILVA residue changes with binding are not conclusive in showing large scale rearrangements of the protein with binding either, but show mostly changes localized to the V domain. The conclusions within the abstract would thus be better toned down slightly to reflect this. One suggestion, the second from last sentence should be changed to, "In particular, a remodeling of electrostatic interactions near the C β H3 helix at the membrane-proximal face of the TCR, a region implicated in interactions with the CD3 co-receptor, suggests a possible role for allostery in TCR signaling." Pointedly, no allosteric mechanism is found within this paper. Other changes within the main text could also reflect this more accurate assessment of the data. I think overall the paper is suggestive of possible V-C connections within TCRbeta upon pMHC binding, but there is no smoking gun here, as it were. As this data will be of interest to the T cell community, publication is recommended, but with the caveats stated in the abstract

RESPONSE TO REVIEWERS

REVIEWERS' COMMENTS:

Reviewer #1 (Remarks to the Author):

The authors have addressed all my concerns and I congratulate them with their achievements. Especially their NMR work is of very high quality and shows the power of solution state methods to address biological aspects in large complexes.

We thank the reviewer for her/his kind remarks on our work.

Reviewer #2 (Remarks to the Author):

The authors have done an excellent job responding to my concerns. This is a real tour de force. My only remaining comment is about relates to the novelty that dynamic allostery resulting from pMHC binding could influence interactions with CD3 proteins. This hypothesis was put forward earlier in Hawse et al. 2013 which showed distributed changes in hydrogen/deuterium exchange upon TCR binding (a possible dynamic allostery mechanism was the title of the paper). The authors do cite this paper, but do not mention it terms of dynamic allostery. This could be fixed by simply citing it in the sentence at the end of the first paragraph on p. 20 ("It is therefore conceivable that p/MHC-induced allosteric...". But great job all around, this will be an important paper that should generate good discussion about signaling mechanisms.

We thank the reviewer for her/his kind remarks on our work.

Although we are not the first to hypothesize a role for allostery in the T cell signaling, we provide additional evidence through NMR chemical shift mapping using both amide and sidechain methyl probes of the TCR/pMHC interaction, as well as provide functional data, in support of that hypothesis. To ensure proper credit is due, we have revised our manuscript to mention the Hawse et al 2013 (ref 42) experiments on pg 20 of the main text to include the term "dynamically driven allostery" as suggested by the reviewer.

Reviewer #3 (Remarks to the Author):

The revised manuscript provides helpful controls, new data and new data presentation that reassure the reviewer that every care has been taken that the results are not due to trivial artifacts or assignment errors. The paper has quite a few new controls and it is convincing that there is little or no self association of the TCRs. They make reasonable arguments that secondary binding sites are not present based on calorimetry and SPR, but they fail to provide a definitive cross saturation experiment with perdeuterated alpha, perdeuterated and ¹⁵N beta, and protonated pMHC. Rather, they show cross saturation data between protonated alpha and perdeuterated ¹⁵N beta to recapitulate the alpha/beta interface and show that alpha does not associate with beta outside of this interface. Technically, the cross saturation experiment is possible if TCRalpha is perdeuterated and otherwise unlabeled, TCRbeta is ¹⁵N/perdeuterated and pMHC is protonated but otherwise unlabeled. Perhaps the size of the complex puts this experiment beyond the authors' capabilities. This omission leaves the possibility of a secondary interaction site between TCRbeta and pMHC open and weakens the paper. As pointed out by the authors, NMR is sensitive to changes not seen by other techniques. This might be just such a case.

In summary, while there is as yet no satisfactory explanation for the magnitude of chemical shift changes, the report appears to show a distal change in the TCRbeta component of TCRalpha-beta with addition of pMHC. Given the lack of mechanism for the distal chemical shift changes and little corroborating evidence in existing paired X-ray structures of liganded and unliganded TCRs, the significance of the isolated chemical shift perturbations is still quite uncertain. The new data showing ILVA residue changes with binding are not conclusive in showing large scale rearrangements of the protein with binding either, but show mostly changes localized to the V domain. The conclusions within the abstract would thus be better toned down slightly to reflect this. One suggestion, the second from last sentence should be changed to, "In particular, a remodeling of electrostatic interactions near the C β H3 helix at the membrane-proximal face of the TCR, a region implicated in interactions with the CD3 co-receptor, suggests a possible role for allostery in TCR signaling." Pointedly, no allosteric mechanism is found within this paper. Other changes within the main text could also reflect this more accurate assessment of the data. I think overall the paper is suggestive of possible V-C connections within TCRbeta upon pMHC binding, but there is no smoking gun here, as it were. As this data will be of interest to the T cell community, publication is recommended, but with the caveats stated in the abstract

We appreciate the reviewer's comments and constructive feedback.

We have toned down the abstract and the main text discussion as suggested by the reviewer.